# Large Language Models are Not Yet Human-Level Evaluators for Abstractive Summarization

**Chenhui Shen**[*][1,2]  **Liying Cheng**[1,3]  **Xuan-Phi Nguyen**[1,3]  **Yang You**[2]  **Lidong Bing**[†][1,3]

[1]DAMO Academy, Alibaba Group, Singapore   [2]National University of Singapore
[3]Hupan Lab, 310023, Hangzhou, China
{chenhui.shen, liying.cheng, x.nguyen, l.bing}@alibaba-inc.com
youy@comp.nus.edu.sg

## Abstract

With the recent undeniable advancement in reasoning abilities in large language models (LLMs) like ChatGPT and GPT-4, there is a growing trend for using LLMs on various tasks. One area where LLMs can be employed is as an alternative evaluation metric for complex generative tasks, which generally demands expensive human judges to complement the traditional automatic metrics for various evaluation dimensions such as fluency and consistency. In this work, we conduct extensive analysis to investigate the stability and reliability of LLMs as automatic evaluators for abstractive summarization. We found that while ChatGPT and GPT-4 outperform the commonly used automatic metrics, they are not ready as human replacements due to significant limitations. That is, LLM evaluators rate each candidate system inconsistently and are dimension-dependent. They also struggle to compare candidates with close performance and become more unreliable with higher-quality summaries by obtaining a lower correlation with humans. In other words, with better abstractive summarization systems being introduced at a fast pace, LLMs may result in misleading and unreliable evaluations.[1]

## 1 Introduction

The desire for inexpensive and fast automatic metrics has never stopped growing. In certain tasks like *extractive* summarization, where full source sentences are selected to appear in the summaries, simple n-gram overlap metrics against the "gold" summaries like ROUGE (Lin, 2004) or BLEU (Papineni et al., 2002) may work well because the correct answer space is narrow. However, for more open tasks like abstractive summarization, there are countless equally good summaries and the "gold" summaries become less important. Although many neural-based metrics such as BERTScore and BARTScore (Zhang et al., 2020b; Yuan et al., 2021), are advocated as more human-aligned, the evaluation criteria are also becoming increasingly complex. As a result, abstractive summarization may not be sufficiently evaluated with automatic metrics (Owczarzak et al., 2012; Nenkova, 2006), and often require extensive human evaluations as complements(Yang et al., 2023; Welbl et al., 2021). However, human evaluations often come with hefty costs and slow iteration cycles, while also being difficult to reproduce and standardize due to small sample sizes and potential human biases (Shen et al., 2022b; Liu et al., 2022).

Recent large language models (LLMs) like ChatGPT and GPT-4 (OpenAI, 2023) have demonstrated outstanding capabilities in language comprehension and reasoning. This leads to a growing trend of employing LLMs as evaluators for complex language generation tasks by prompting them with carefully and elaborately crafted instructions (Chiang and Lee, 2023; Gao et al., 2023; Wang et al., 2023a; Wu et al., 2023; Luo et al., 2023; Liu et al., 2023). Despite the preliminary success suggested by such works, it is still inconclusive as to what degree of confidence we can trust the evaluation results produced by LLMs across different dimensions, despite their supposedly high average correlation with humans. It is also unclear if certain LLM-based metrics are more reliable than others, or if their reliability and fairness vary for different candidate systems.

In this work, we conduct extensive analysis to assess whether LLM evaluators can reliably replace human judges. Specifically, we incorporate two common human evaluation approaches with LLM evaluators, namely Likert-scale scoring (He et al., 2022; Shen et al., 2022b; Zhang et al., 2020a) and head-to-head (H2H) comparisons (Shen et al., 2022a; Li et al., 2020; Liu and Lapata, 2019). For

---

[*]Chenhui is under the Joint PhD Program between Alibaba and National University of Singapore.
[†] Corresponding author.

[1]Our code and data are fully released at https://github.com/DAMO-NLP-SG/LLM_summeval.

Likert-scale scoring, we explore direct reason-then-score (RTS) generation and a multiple-choice question (MCQ) method. The former instructs the LLM to provide reasoning before giving a score, while the latter simply prompts it to choose a specific score with a pre-determined description as the reason. For the Head-to-Head (H2H) comparison, we prompt LLM for a preference over the summaries from two compared candidate systems.

Our experiments show that LLM evaluators, with RTS and MCQ, outperform existing automatic metrics (Lin, 2004; Yuan et al., 2021). However, they are **not** ready to be reliable alternatives for human evaluation yet. Specifically, (*i*) LLM evaluators struggle to distinguish candidates with close performances (§ 4.2.1). (*ii*) LLM evaluators are candidate-dependent, meaning they do not exhibit highly consistent degrees of human alignment for different candidates (§ 4.2.3). Thus, they may unfairly favor or disfavor an evaluated candidate. (*iii*) LLM evaluators are dimension-dependent, meaning they have varying degrees of evaluation capabilities for different dimensions like coherence and fluency (§ 4.2.3). (*iv*) Lastly, as the quality of summaries improves with better candidates, LLM evaluators become unreliably less correlated with human judgments, according to our newly proposed meta-correlation metric (§ 4.2.4).

While we still call for a better automatic metric, in the meantime, we suggest a temporary solution in § 5 for abstractive summarization practitioners to use LLMs more reliably. Specifically, we advocate calculating the correlation between RTS and MCQ as a preliminary indicator of the reliability of the LLM for certain dimensions. If RTS and MCQ do not generally agree with each other, then further human evaluations are required.

## 2 Related Work

**Summarization** The summarization task involves generating a summary that contains concise and important (i.e., salient) contents of the original input article (Nenkova and McKeown, 2012). This task has been handled with 2 different approaches: extractive and abstractive. Unlike extractive summarization systems that directly extract salient phrases or sentences from the input article (Ernst et al., 2022; Chen et al., 2021; Zhou et al., 2018; Dong et al., 2018), abstractive summarization systems are expected to generate summaries using their own words and apply sentence fusion

or paraphrasing techniques (Shen et al., 2023; Liu et al., 2022; Xiao et al., 2022; Lewis et al., 2020; Zhang et al., 2020a; Ziegler et al., 2019; Bing et al., 2015; Xu and Durrett, 2021). As such, abstractive summarization poses significantly more challenges for automatic and human evaluation pipelines (Saha et al., 2022; Pagnoni et al., 2021), because it is increasingly insufficient to use the provided "gold" summary as ground truth.

**Human Evaluation** Human evaluation can be conducted with different approaches. Some work (He et al., 2022; Shen et al., 2022b; Zhang et al., 2020a; Cheng et al., 2020; Gao et al., 2019; Liu et al., 2018; Li et al., 2017; Kryściński et al., 2018) employ a Likert scale to evaluate the summaries on discrete ranges, such as from 1 to 5. Meanwhile, many others suggest comparison approaches by asking human annotators to select the best summary out of 2 or more generated summaries from different systems (Shen et al., 2022a; Li et al., 2020; Liu and Lapata, 2019; Fan et al., 2018; Fabbri et al., 2019). Following this, we test LLM-based evaluators using both approaches with human-friendly instruction prompts.

**Automatic Evaluation** ROUGE (Lin, 2004) has been a common lexical overlap metric to evaluate summarization systems. Apparently, ROUGE is not sufficient for abstractive summarization, because the "gold" labels it relies on cannot comprehensively account for the complexity and variability of this task. In addition, the common usage of sentence fusion techniques and novel words for abstractive summarization may make ROUGE even less reliable. Zhang et al. (2020b) propose the neural-based BERTScore, which leverages the BERT word embeddings to compute the semantic similarity among tokens. Yuan et al. (2021) later introduce BARTScore, which uses BART (Lewis et al., 2020) to compute the probability of a summary given its input article. Nonetheless, these metrics may not reflect all of the complicated evaluation dimensions required for abstractive summarization mentioned earlier, nor do they have sufficiently high correlations with humans.

**LLM-based Evaluation** There are many concurrent works that demonstrate the potential of LLMs to conduct complex human tasks (Chiang and Lee, 2023; Gao et al., 2023; Wang et al., 2023a; Wu et al., 2023; Luo et al., 2023; Liu et al., 2023; Cheng et al., 2023). The key advantage of instruction-

tuned LLMs, like ChatGPT or GPT-4 (Ouyang et al., 2022; OpenAI, 2023), is that we can explicitly describe in natural language what our evaluation criteria and dimensions are and how to score the summaries, similar to how we would explain such tasks to a human expert. Chiang and Lee (2023) use LLMs for open-ended story evaluations, while Luo et al. (2023) apply ChatGPT specifically for evaluating the consistency of summaries. Wu et al. (2023) formulate LLMs as diverse role-players to evaluate summaries from the perspectives of different personas. Wang et al. (2023a) and Liu et al. (2023) also explore the LLM's evaluation potential in various dimensions for the natural language generation task. Our work differs from the above works in that besides investigating the LLMs' capability using different approaches across various dimensions for abstractive summarization, we further focus on the *reliability* of LLM across evaluated systems and dimensions.

## 3 LLM as a Zero-Shot Evaluator

We investigate an LLM's evaluation capabilities in the dimensions of coherence, consistency, fluency, and relevance respectively, as defined by Fabbri et al. (2021) (see Appendix A). Following common human evaluation approaches, we propose two methods for Likert-scale scoring, namely the reason-then-score method and the multiple-choice question method, as well as one method for head-to-head comparisons. We describe each method in § 3.1 using the relevance dimension as an example (see more prompts and details in Appendix B). We further experiment with alternative phrasings for different methods in Appendix C.

Besides exploring different evaluation methods, the stability of LLM-based evaluations across different summarization systems is equally important. Ideally, a stable LLM evaluator should perform equally well regardless of the evaluated systems, with a close (if not identical) degree of alignment with human judgments. In § 3.2, we propose a meta-correlation metric and explain how it can gauge the extent to which LLM evaluators' performances depend on the evaluated systems, which indicates how stable and reliable they may be with evaluating any future candidate systems.

### 3.1 Summary Evaluation Methods

**Reason-then-Score (RTS)**   Given the success of chain-of-thought prompting (Kojima et al., 2022;

---

Score the following Summary given the corresponding Article with respect to relevance from one to five, where one indicates "irrelevance", and five indicates "perfect relevance". Note that relevance measures the Summary's selection of important content from the Article, whether the Summary grasps the main message of the Article without being overwhelmed by unnecessary or less significant details.

Article: {article}

Summary: {summary}

Provide your reason in one sentence, then give a final score:

---

Table 1: Example prompt for the RTS method on the relevance dimension. Texts in {blue} represent the article and the corresponding summary to be evaluated.

---

Choose an option from A to E in order to score the following Summary given the corresponding Article with respect to relevance from one to five, where one indicates "irrelevance", and five indicates "perfect relevance". Note that relevance measures the Summary's selection of important content from the Article, whether the Summary grasps the main message of the Article without being overwhelmed by unnecessary or less significant details.

Article: {article}

Summary: {summary}

A: The Summary is totally irrelevant to the Article. Score: One.
B: The majority of the Summary is irrelevant to the Article. Score: Two.
C: Some information in the Summary is relevant to the Article whereas some are not. Score: Three.
D: The majority of the Summary is relevant to the Article. Score: Four.
E: All information included in the Summary is relevant to the Article. Score: Five.

Your Answer (enter 1 letter from A to E):

---

Table 2: Example prompt for the MCQ method on the relevance dimension. Texts in {blue} represent the article and the corresponding summary to be evaluated.

Wei et al., 2022), an intuitive method is to ask the LLM to evaluate a specific dimension by first generating the reasoning and then a corresponding score. Since the SummEval dataset (Fabbri et al., 2021) contains human scores on a Likert scale of 1 to 5, we also ask the LLM to score the summaries in the same range, as shown in Table 1.

**MCQ Scoring (MCQ)**   Nevertheless, previous works find that the reasoning generated by the LLM does not always make sense (Lyu et al., 2023;

Article: {article}

Summary #1: {summary from model A}

Summary #2: {summary from model B}

A: Summary #1 is more relevant.
B: Summary #2 is more relevant.
C: Both Summary #1 and Summary #2 are equally relevant.

Table 3: Example prompt for the H2H method on the relevance dimension. Text in {blue}: the specific article, and the corresponding summaries generated by a pair of compared models.

Wang et al., 2023b; Gao et al., 2022). To avoid the misguidance of wrongly generated reasoning, we explore a more constrained MCQ method for the Likert-scale scoring. As shown in Table 2, instead of allowing the LLM to freely generate its thoughts, we dictate specific reasoning for each score.

**Head-to-Head Comparison (H2H)** Lastly, some concurrent works also observe that ChatGPT can act as an effective ranking model (Ma et al., 2023a,b). We thus explore the head-to-head comparison approach for LLM-based evaluations. As shown in Table 3, we present 2 summaries (Summary #1 and #2) generated by different summarization systems on the same input article, then prompt the LLM to select the better summary, or to indicate a tie. Moreover, to avoid potential biases that arise from the summary IDs, we conduct each evaluation twice, presenting the same summary as either #1 or #2 respectively.

## 3.2 Stability of LLM Evaluators

To ensure fairness across all evaluated systems, we argue that it is crucial for LLMs to produce stable evaluations. That is, regardless of evaluated systems, the LLMs should maintain a consistent degree of alignment with human judgments. We investigate such stability in two ways.

First, We categorize the summaries based on their originating summarization systems, and then examine the correlation between the LLM and human evaluations for each system. Ideally, if an LLM is stable across systems, it should produce evaluations that are similarly correlated to human evaluations. Otherwise, if the correlations differ significantly across different candidates, then we may conclude that the LLM's evaluations are system-dependent.

Second, we define a **meta-correlation** metric to quantify the extent to which the LLM's performance is affected by the quality of the evaluated systems. Specifically, we use the average human score for each candidate as an indicator of its summarization quality ($Q_i$), as shown in Equation (1):

$$Q_i = \frac{1}{N} \sum_{j=1}^{N} f_{\text{human}}(g_{i,j}) \qquad (1)$$

where $f_{\text{human}}(\cdot)$ indicates the human evaluation, $g_{i,j}$ represents the $j^{th}$ summary generated by the $i^{th}$ candidate system. Each candidate's quality is calculated as an average of $N$ generated summaries ($N = 100$ for all systems). Next, we use the correlation $P_i$ between LLM scores and human scores as an indicator of the LLM's evaluation performance for the $i^{th}$ candidate, as follows:

$$P_i = \rho([f_{\text{LLM}}(g_{i,1}), ..., f_{\text{LLM}}(g_{i,N})], \\ [f_{\text{human}}(g_{i,1}), ..., f_{\text{human}}(g_{i,N})]) \qquad (2)$$

where $\rho$ denotes the correlation metric (*i.e.,* Spearman correlation, Pearson correlation, or Kendall's Tau[2]), and $f_{\text{LLM}}(\cdot)$ indicates the LLM's evaluation for each summary $g_{i,j}$. Finally, we calculate the meta-correlation[3] $M$ on a total of $k$ candidates as:

$$M = \rho([Q_1, ..., Q_k], [P_1, ..., P_k]) \qquad (3)$$

Ideally, an LLM should work well regardless of the quality of the evaluated systems, which means that $M$ should be close to zero. On the other hand, a significant $M$ would indicate an undesirable relationship between the LLM's evaluation capability and the quality of the evaluated systems, suggesting that the LLM evaluation is not stable, such that it may not evaluate each candidate system fairly using the same standards.

---

[2]We use the scipy.stats.kendalltau package, which implements the tau-b variant that accounts for ties.

[3]We use the same correlation metric for both Equation2 and 3. For instance, if $P_i$ is obtained using Spearman correlation, then $M$ is also calculated using Spearman correlation.

# 4 Experiments

## 4.1 Setups

We use the ChatGPT "gpt-3.5-turbo-0301" snapshot (§ 4.2) for all three methods. By using a fixed snapshot, we ensure all evaluations are conducted with the same LLM model. In addition, we evaluate with the GPT-4 "gpt-4-0314" snapshot (§ 4.3) using the best evaluation method determined by ChatGPT to check for any potential improvement. Given that ChatGPT and GPT-4 are amongst the top performing LLMs, we use their performance to estimate the potential of LLMs as reliable evaluators. Additional results using three different-sized Llama 2 models (Touvron et al., 2023) are reported in Appendix D, which all performs worse. Similar to Luo et al. (2023) and Wu et al. (2023), we set the temperature to 0 and reset the dialogue history for each evaluation instance.

**Dataset** We use the SummEval benchmark dataset (Fabbri et al., 2021). This dataset contains expert human annotations for coherence, consistency, fluency, and relevance on the generation results from 12 abstractive systems (see details in Appendix table 21) on the CNN/DM dataset (Hermann et al., 2015). Each evaluated system generates summaries for the same 100 news articles, and each summary is scored by 3 expert annotators from 1 to 5. The annotations achieve with a high kappa coefficient of 0.713 (Fabbri et al., 2021). We further calculate the annotations' standard deviations across each evaluated system in Appendix Table 20. Given a step size of 1, the standard deviations are considered very small, thus suggesting that this dataset has a high level of human agreement. Following Chiang and Lee (2023), Chhun et al. (2022), and Guan and Huang (2020), we use the average human scores as the reference scores.

**Baselines** We use ROUGE (Lin, 2004) F1 scores for ROUGE-1, ROUGE-2, and ROUGE-L, BERTScore (Zhang et al., 2020b), BARTScore, BARTScore-CNN, and BARTScore-CNN-PARA (Yuan et al., 2021) as baseline metrics. The last two metrics use BART models fine-tuned on CNN/DM's training data, and are especially strong.

**Prompts** We conduct evaluation following our prompt formats given in Table 1, 2, and 3. Following Fabbri et al. (2021), we re-use the definitions of the evaluation dimensions: (*i*) Coherence - the collective quality of all sentences, (*ii*) Consistency - the factual alignment between the summary and the summarized source, (*iii*) Fluency - the quality of individual sentences, and (*iv*) Relevance - the selection of important content from the source.

**Measurements** To compare all evaluation methods on equal ground with human evaluation, we use four different measurements. First, we count the number of *correct preferences* (#CP), which is the number of times each automatic metric has the same preference as the average human scores do over a set of compared system pairs (§ 4.2.1). This can help measure the alignment of evaluation methods with humans at a granular level. To determine the preferred system by a particular metric, we assign a system 1 point if its generated summary is evaluated as better than that of the other system according to the metric, or assign both systems 0.5 for a tie. Then, we aggregate the different scores for the compared systems for all 100 test inputs, and the system with a higher score is considered the preferred system by that metric (see Appendix Table 22 for details).

Next, we also use the *Pearson* correlation (Cohen et al., 2009), *Spearman* correlation (Spearman, 1987), and *Kendall's* Tau (Kendall, 1938) to measure the relationship between the scores of automatic evaluators and humans (§ 4.2.2, 4.2.3, 4.2.4). While the Pearson score measures linear relationships, the other two measure the ordinal relationship that may be non-linear. Moreover, Kendall's Tau is less sensitive than Spearman correlation to outliers due to its paired counting of concordant and discordant pairs.

## 4.2 ChatGPT Evaluator

In this section, we examine the ChatGPT evaluator across many aspects, ranging from human correlation and stability across different systems.

### 4.2.1 Correct Preferences

The ultimate goal of evaluation is to determine if one candidate system is better than the other in a compared pair. The number of correct preferences (#CP) metric normalizes all evaluation methods into determining whether an evaluator can, as a human expert would, pick the same better system or determine a tie. We conduct such analysis with different pairs of summarization systems on the same input articles. Due to the limited budget for API calls, we only evaluate H2H on a challenge set, consisting of 11 candidate pairs with the closest

| Metrics | Coherence | | | Consistency | | | Fluency | | | Relevance | | |
|---|---|---|---|---|---|---|---|---|---|---|---|---|
| | Spear. | Pear. | Kend. | Spear. | Pear. | Kend. | Spear. | Pear. | Kend. | Spear. | Pear. | Kend. |
| ROUGE-1* | 0.193 | 0.202 | 0.136 | 0.155 | 0.186 | 0.121 | 0.075 | 0.153 | 0.058 | 0.323 | 0.361 | 0.231 |
| ROUGE-2* | 0.145 | 0.146 | 0.101 | 0.137 | 0.156 | 0.107 | 0.053 | 0.095 | 0.041 | 0.255 | 0.262 | 0.181 |
| ROUGE-L* | 0.148 | 0.158 | 0.105 | 0.133 | 0.167 | 0.103 | 0.078 | 0.146 | 0.060 | 0.306 | 0.340 | 0.219 |
| BERTScore* | 0.375 | 0.383 | 0.265 | 0.163 | 0.182 | 0.127 | 0.167 | 0.229 | 0.130 | 0.396 | 0.414 | 0.285 |
| BARTScore* | 0.381 | 0.391 | 0.275 | 0.271 | 0.265 | 0.212 | 0.168 | 0.187 | 0.131 | 0.381 | 0.391 | 0.276 |
| BARTScore-CNN* | **0.461** | **0.480** | 0.332 | 0.389 | 0.413 | 0.305 | 0.310 | 0.378 | 0.241 | 0.425 | 0.450 | 0.309 |
| BARTScore-CNN-PARA* | 0.455 | 0.455 | 0.328 | 0.413 | 0.459 | 0.324 | 0.368 | 0.417 | 0.286 | 0.414 | 0.440 | 0.299 |
| ChatGPT-RTS | 0.388 | 0.399 | 0.312 | 0.423 | 0.532 | 0.378 | 0.285 | 0.302 | 0.240 | **0.448** | **0.463** | 0.357 |
| ChatGPT-MCQ | 0.424 | 0.416 | 0.350 | 0.343 | 0.487 | 0.320 | 0.343 | 0.431 | 0.305 | 0.384 | 0.395 | 0.329 |
| GPT-4-RTS | 0.427 | 0.461 | **0.361** | **0.556** | **0.618** | **0.522** | **0.498** | **0.600** | **0.452** | **0.448** | 0.428 | **0.373** |

Table 4: Spearman (Spear.) correlations, Pearson (Pear.) correlations, and Kendall's Tau (Kend.) between various metrics and human scores for a total of 1,200 summaries. *: results derived from Wang et al. (2023a). **Bolded**: best results. Underlined: second best results. Values in light gray color are insignificant (p-value $\geq$ 0.05).

| Metrics | Coh | Con | Flu | Rel | Avg |
|---|---|---|---|---|---|
| Random | 3.67/22 | 3.67/22 | 3.67/22 | 3.67/22 | 3.67/22 |
| ROUGE-1 | 3/40 | 5/46 | 3/46 | 4/47 | 3.75/44.75 |
| ROUGE-2 | 2/38 | 7/48 | 3/45 | 4/46 | 4.00/44.25 |
| ROUGE-L | 2/31 | 5/37 | 4/39 | 6/41 | 4.25/37.00 |
| BERTScore | 4/48 | 5/44 | 4/44 | 6/46 | 4.75/45.50 |
| BARTScore | 8/46 | 7/48 | 5/45 | 6/46 | 6.50/46.25 |
| BARTScore-CNN | **9**/53 | 5/53 | 5/54 | 4/53 | 5.75/53.25 |
| BARTScore-CNN-PARA | **9**/49 | **8**/54 | 6/52 | 5/51 | **7.00**/51.50 |
| ChatGPT-RTS | 6/**54** | 6/**56** | **9**/**62** | 7/**62** | **7.00**/**58.50** |
| ChatGPT-MCQ | 5/**54** | 7/**56** | 8/60 | 7/58 | 6.75/57.00 |
| ChatGPT-H2H | 8/- | 7/- | 7/- | 4/- | 6.50/- |
| GPT-4-RTS | 5/55 | 7/53 | 8/60 | 7/56 | 6.75/56.00 |

Table 5: Number of correct preferences (#CP) on the 11-pair challenge set (in black) and the 66-pair full set (in brown). Random: for each pair, there are three possibilities (two possibilities for one model being better, one possibility for a tie) so the random #CP is one-third of the total compared pairs.

average performances according to human scores. However, for RTS, MCQ, and other baselines, we can easily calculate the #CP for all 66 possible pairs (see Appendix E).

Table 5 reports the #CP for both the standard 66-pair full set (in brown) and the 11-pair challenge set (in black). As shown for the larger standard set, RTS unanimously obtains the largest #CP across all dimensions, with an average of 58.5 out of 66 candidate pairs (i.e. 88.6% accuracy).

Despite the high overall accuracy, weaknesses of such evaluators are revealed as we dive into their performances in the 11-pair challenge set (black scores of Table 5), where the evaluated candidates are close matches. Specifically, BARTScore-CNN-Para performs better than RTS in coherence and consistency, possibly because it is fine-tuned with same-domain summarization data. For fluency and relevance, ChatGPT-RTS still performs best among all evaluators. Nonetheless, its average accuracy

drops significantly to 63.6% (7 out of 11), which indicates LLM evaluators struggle to differentiate the closely matched candidate systems. In other words, LLM evaluators may only reliably compare candidates with a relatively large performance gap.

### 4.2.2 Correlations with Human

Table 4 reports that Spearman, Pearson correlations, and Kendall's Tau between scores of multiple automatic evaluators and humans with a total of 1200 summaries from all systems, across the four evaluation dimensions. As shown, ChatGPT RTS and MCQ demonstrate stronger correlations with humans than many automatic evaluators, such as ROUGE and BARTScore, with up to 0.2 gains in fluency. While RTS achieves higher correlations in the dimensions of consistency and relevance, MCQ has relatively strong correlations in the dimensions of coherence and fluency. Meanwhile, the specialized BARTScore-CNN family also shows competitive performance in coherence, most likely due to the fine-tuning process with CNN/DM.

### 4.2.3 Per-candidate Correlations

Next, we break down the human correlation of ChatGPT-RTS for each candidate system and measure the statistical spread for the correlations across all systems (see raw results in Appendix table 23). Ideally, a stable evaluator should exhibit the same human correlation across candidates and dimensions, and display flattened boxes in a line.

However, as illustrated in Figure 1, the spread of correlations for different candidates is particularly wide, with up to 0.5 correlation difference in consistency. This means that the RTS evaluator exhibits a significantly varying degree of alignment with human judgment for different candidates. In other words, ChatGPT-RTS is *candidate-dependent*

| | Coherence | | | Consistency | | | Fluency | | | Relevance | | |
|---|---|---|---|---|---|---|---|---|---|---|---|---|
| | Spear. | Pear. | Kend. | Spear. | Pear. | Kend. | Spear. | Pear. | Kend. | Spear. | Pear. | Kend. |
| ROUGE-1 | 0.0 | -0.032 | -0.091 | -0.490 | -0.527 | -0.303 | -0.420 | -0.518 | -0.273 | -0.420 | -0.387 | -0.273 |
| ROUGE-2 | 0.559 | 0.508 | 0.485 | -0.259 | -0.480 | -0.152 | -0.217 | -0.438 | -0.152 | -0.084 | -0.120 | -0.121 |
| ROUGE-L | 0.231 | 0.251 | 0.121 | -0.510 | -0.522 | -0.303 | -0.168 | -0.412 | -0.152 | -0.224 | -0.266 | -0.121 |
| BERTScore | -0.413 | -0.403 | -0.212 | -0.580 | -0.869 | -0.424 | -0.455 | -0.663 | -0.303 | -0.685 | -0.756 | -0.515 |
| BARTScore | **-0.916** | -0.747 | **-0.788** | -0.266 | -0.504 | -0.121 | 0.154 | 0.123 | 0.182 | -0.769 | -0.837 | -0.606 |
| BARTScore-CNN | -0.748 | -0.800 | -0.636 | -0.671 | **-0.913** | -0.515 | -0.510 | -0.604 | -0.485 | -0.825 | -0.852 | -0.667 |
| BARTScore-CNN-PARA | -0.720 | **-0.858** | -0.606 | -0.685 | -0.888 | -0.576 | -0.294 | -0.522 | -0.212 | -0.853 | **-0.880** | -0.727 |
| ChatGPT-RTS | -0.042 | -0.072 | -0.121 | -0.811 | -0.751 | **-0.636** | -0.748 | **-0.728** | **-0.606** | -0.559 | -0.473 | -0.394 |
| ChatGPT-MCQ | -0.175 | -0.11 | -0.182 | **-0.818** | -0.411 | **-0.636** | -0.622 | -0.484 | -0.394 | -0.350 | -0.622 | -0.212 |
| GPT-4-RTS | -0.531 | -0.678 | -0.424 | -0.600 | -0.103 | -0.236 | **-0.839** | -0.520 | -0.515 | **-0.958** | **-0.880** | **-0.848** |

Table 6: Meta-correlation for various evaluation methods. **Bolded**: most negative meta-correlation. Underlined: second most negative meta-correlation. Values in light gray color are insignificant (p-value $\geq 0.05$).

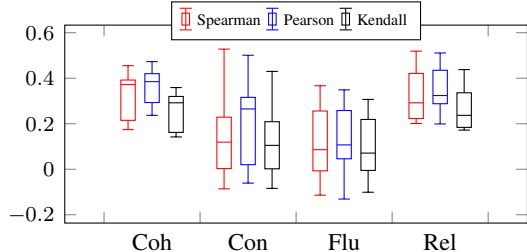

Figure 1: Spread of per-candidate correlations with human scores for ChatGPT-RTS evaluations.

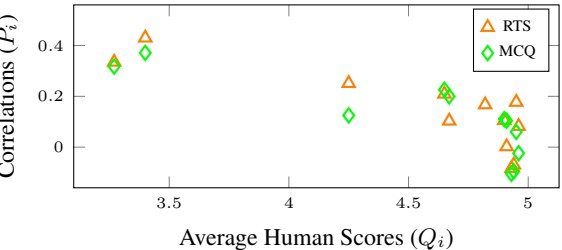

Figure 3: Relationship between per-model correlations (Kendall's Tau) and human scores on consistency.

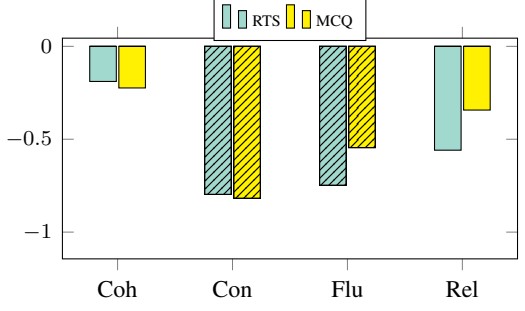

Figure 2: Meta-correlation (Kendall's Tau) for RTS and MCQ. Shaded: statistically significant with $p < 0.05$.

and one should not expect such LLM evaluators to have the same level of human alignment on a new summarization system. Similar trends can also be observed for MCQ (see Appendix table 24).

In addition, the medians across the four dimensions are also different. This indicates that the ChatGPT is also *dimension-dependent* and unstable. Given such varying performances across different dimensions, ChatGPT may not behave well with a newly introduced evaluation criterion.

### 4.2.4 Summary Quality *vs* Human Alignment

Using our proposed meta-correlation measurement in § 3.2, we analyze the relationship between summary quality and human correlation of LLM evaluators. We illustrate the meta-correlation in terms of

Kendall's Tau for both RTS and MCQ in Figure 2. As shown, both RTS and MCQ exhibit strong negative meta-correlation for consistency and fluency. This suggests that ChatGPT becomes less human-aligned with improving qualities of the evaluated systems.

To illustrate this phenomenon further, we scatter the paired coordinates of the summarization system quality ($Q_i$, Equation (1)) and ChatGPT's evaluation performance ($P_i$, Equation (2)) in Figure 3. As shown, while the LLM evaluator is better human-correlated with lower-quality candidates ($< 3.5$), it is less reliable when dealing with high-quality candidates ($> 4.7$) with much lower and inconsistent correlations.

We compare the meta-correlation for all evaluation metrics in Table 6. We can see that while the ROUGE metrics exhibit no significantly negative meta-correlation, the neural metrics all display significant meta-correlation in certain dimensions. One highly likely reason for this behavior is due to the varying biases inherent to the neural models, which would explain why ROUGE as a simple n-gram overlap metric doesn't exhibit significant negative meta-correlations. Interestingly, ROUGE-2 even shows a strong positive meta-correlation on coherence (which is plausible, because bi-gram

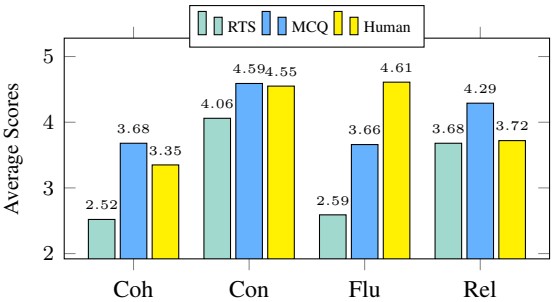

Figure 4: The average ChatGPT RTS and MCQ scores and human scores across dimensions.

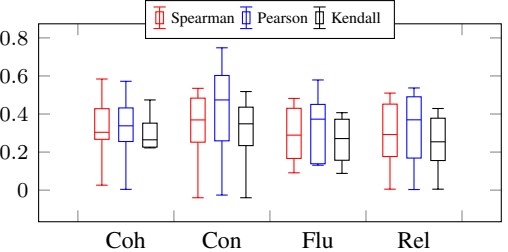

Figure 5: GPT-4's (RTS) spread of per-candidate correlations.

overlap performance may be more accurate as candidates produce more coherent texts).

Both the BARTScore variants and LLMs demonstrate the most negative meta-correlations. ChatGPT-RTS has the most negative meta-correlation in the dimensions of consistency and fluency, indicating that it may be the least reliable to evaluate high-quality systems on these dimensions. On the other hand, the BARTScore family may be unreliable in comparing systems with high qualities of coherence, consistency, and relevance.

So far, the observations discussed in § 4.2.3 and § 4.2.4 collectively suggest that LLM evaluators may not be a reliable standalone metric for challenging scenarios, and further human evaluation is required for conclusive decisions.

### 4.2.5 RTS and MCQ Scores

Lastly, we delve into the detailed scores generated by ChatGPT with either the RTS or MCQ method. Since both methods score the summaries in the same range of human scores of 1 to 5 (Fabbri et al., 2021), we can show a direct comparison of the average RTS and MCQ scores with human scores in Figure 4 (see more details in Appendix F). As shown, the RTS scores are much lower than the human scores across all dimensions, while MCQ scores are consistently higher and better match the human scores (except for relevance). In other words, while RTS is best aligned with humans according to § 4.2.1 and § 4.2.2, we cannot replace the human scores with RTS scores in absolute terms.

The discrepancy may be attributed to the unfaithful reasoning generated by LLMs (Lyu et al., 2023; Wang et al., 2023b; Gao et al., 2022). Our further investigation suggests that ChatGPT-RTS generates false or unrelated-to-dimension reasoning. Thus, it is possible that the much lower scores could be caused by ChatGPT penalizing the sum-

maries according to false premises (more examples in Appendix G). For instance, RTS may penalize the summary's repetitiveness in the consistency dimension or suppress fluency ratings for missing important details.[4] On the other hand, the MCQ counterpart gives higher overall scores, most likely because the confined set of pre-defined reasons prevents such unrelated penalization, though not leading to better human alignment.

### 4.3 GPT-4 Evaluator

A natural question to ask is whether such aforementioned limitations are resolved with an stronger LLM. In this section, we conduct similar analyses on GPT-4 (OpenAI, 2023) with the RTS method. We present the GPT-4 results in the last rows of Table 4 and 5. The results suggest that a stronger LLM does not necessarily translate to a stronger LLM evaluator, although Table 4 does show that GPT-4 outperforms ChatGPT in terms of human correlation consistently across most dimensions.

Unfortunately, GPT-4 still suffers from the same limitations as ChatGPT. It appears to be both candidate-dependent and dimension-dependent, as demonstrated by the large spreads with varying median values across dimensions in Figure 5 and the significantly negative meta-correlations out of 3 dimensions (Table 6). However, GPT-4 is less dimension-dependant as compared to ChatGPT, as the medians in the box plots in Figure 5 are more aligned than those in Figure 1.

In addition, there is a notable enhancement in the meta-correlation for consistency, which we attribute to a significant reduction in reported hallucinations with GPT-4 (OpenAI, 2023). It is possible that with much more instruction training to avoid hallucinations, GPT-4 is much better aligned with humans to detect inconsistencies (i.e. hallucina-

---

[4]Fabbri et al. (2021) observe similar issues with crowd-sourced non-expert annotators.

| | Coherence | | | Consistency | | | Fluency | | | Relevance | | |
|---|---|---|---|---|---|---|---|---|---|---|---|---|
| | Spear. | Pear. | Kend. | Spear. | Pear. | Kend. | Spear. | Pear. | Kend. | Spear. | Pear. | Kend. |
| $\rho(R_i, P_i^{\text{RTS}})$ | 0.343 | 0.198 | 0.091 | 0.685 | 0.506 | 0.576 | 0.727 | 0.797 | 0.545 | 0.168 | 0.128 | 0.000 |
| $\rho(R_i, P_i^{\text{MCQ}})$ | 0.657 | 0.625 | 0.394 | 0.685 | 0.616 | 0.515 | 0.322 | 0.573 | 0.212 | 0.091 | -0.106 | 0.000 |

Table 7: Correlations between RTS-MCQ $R_i$ and RTS-Human ($P_i^{\text{RTS}}$) and MCQ-Human ($P_i^{\text{MCQ}}$). High values suggest $R_i$ can be a reliability indicator for RTS and MCQ. Light gray values are insignificant (p $\geq$ 0.05).

tions) in summaries.

Nevertheless, GPT-4 exhibits a much worse negative meta-correlation in the relevance dimension, which, interestingly, seems to reflect the challenges of maintaining both "truthfulness" and "informativeness" (Ouyang et al., 2022). This is because a model could be easily made more truthful if allowed to provide less relevant information (for instance, by refusing to answer the users' questions). It is possible that with reduced capability in the informativeness dimension, the model is less capable of differentiating the nuances of less relevant summaries when the summary quality is generally high. Nevertheless, we leave it to future work to determine whether GPT-4's more negative meta-correlation in the relevance dimension could be related to its stronger performance in consistency. We provide more details on the GPT-4 evaluator in Appendix H.

## 5 A Temporary Efficient Framework

Despite the aforementioned limitations, it may be hard to resist the temptation of using LLM evaluators given their superiority over other automatic metrics. In such a case, one should be able to tell when LLM evaluators are more likely to be unreliable and employ further human evaluation when necessary. To this end, we suggest combining the RTS and MCQ scores as a cost-efficient framework. Specifically, we calculate the correlation between RTS and MCQ scores for the $i^{th}$ candidate system as a reliability indicator:

$$R_i = \rho([f_{\text{RTS}}(g_{i,1}), ..., f_{\text{RTS}}(g_{i,N})], \\ [f_{\text{MCQ}}(g_{i,1}), ..., f_{\text{MCQ}}(g_{i,N})]) \quad (4)$$

Then, we can loosely infer that up to a reliability tolerance $r \in (0, 1)$, the LLM evaluators (either RTS or MCQ) are reliable if $R_i > r$. In other words, given a candidate $i$, if RTS and MCQ agree with each other up to a certain degree of tolerance $r$, we may assume the evaluator is reliable enough to avoid invoking further human evaluation.

To validate this theory, we measure the correlations $\rho(R_i, P_i^{\text{RTS}})$ or $\rho(R_i, P_i^{\text{MCQ}})$, where

$P_i^{RTS/MCQ}$ is the performance of either method as defined in Equation (2). Given significantly large positive values of either $\rho(R_i, P_i^{\text{RTS}})$ or $\rho(R_i, P_i^{\text{MCQ}})$, we can then conclude that $R_i$ can be used as a reliable indicator for the performance of the corresponding method.

As shown in Table 7, $R_i$ demonstrates a significant correlation with $P_i^{RTS}$ on both the consistency and fluency dimensions, and with $P_i^{MCQ}$ on the coherence and consistency dimensions. This means that if RTS and MCQ generally agree with each other on the candidate's performance on a particular dimension with high $\rho(R_i, P_i^{\text{RTS}})$ (or $\rho(R_i, P_i^{\text{MCQ}})$), RTS (or MCQ) is more likely to be human-aligned. Meanwhile, if RTS disagrees with MCQ ($R_i < r$), further human evaluators are required to provide a conclusive evaluation. We provide $R_i$ values for ChatGPT on each evaluated system in Appendix Table 29.

## 6 Conclusion

We explore the potential of using LLMs with different prompting techniques as metrics for abstractive summarization systems. Our extensive analysis suggests that while LLMs like ChatGPT perform better than commonly used automatic metrics across different summarization systems and dimensions, they are still not ready to replace human evaluators because they are candidate- and dimension-dependent, and they do not align well with human when comparing high-quality candidates. Nonetheless, if an LLM evaluator is to be used, we suggest combining multiple evaluation methods as a preliminary indicator to determine whether the metric is likely to be unreliable and whether further human evaluation is required.

## Limitations

**Potential Human Bias.** We benchmark the LLM evaluation results against the average of three human expert scores. Naturally, it is possible that these scores may exhibit potential biases of the human experts. Nevertheless, we wish to explore

whether LLM evaluators are aligned with human experts, and may naturally exhibit the same bias as a human would. In other words, we examine whether we can reliably replace human annotators with LLMs, instead of seeking a "perfect" solution that has absolutely zero bias.

**Dataset Size.** Given the constraints of the small size of the human-annotated SummEval dataset, we could only evaluate 100 summaries generated for each summarization system, with a total of 12 abstractive summarization systems. Since we have observed a significant correlation of LLM evaluations with humans for the consolidated 1200 summaries across all systems, it is possible that with a larger evaluation number, the per-system correlation could also be improved. In addition, given only 12 evaluated systems, our meta-correlation may still be subject to sample biases. We leave more investigations for the future once there are larger annotated datasets.

**Prompt tuning.** Designing better prompt for LLMs are also ongoing research. Although it is possible that LLMs may act as better evaluators with better prompts, prompt tuning is not our focus. We seek to highlight the limitations of the investigated LLMs and have demonstrated that limitations such as negative meta-correlation are also found with a few other alternative prompts (see Appendix C).

**Availability of Commercialized LLM** We note that the "gpt-3.5-turbo-0301" snapshot is currently taken down[5] by OpenAI and replaced with a newer snapshot, "gpt-3.5-turbo-0613". This is also one disadvantage of using out-of-the-box commercialized LLM for summarization evaluations, as the exact checkpoints may not be stably available. As a result, future models may not be fairly compared against previously evaluated models using a different LLM checkpoint. Nevertheless, our paper only seeks to investigate the potential of LLM as an out-of-the-box evaluator, and the OpenAI models are currently one of the strongest. Eventually, we wish to raise awareness of some of the significant limitations found with these LLMs, which need to be resolved before LLMs can be used as direct replacements for human evaluations. In addition, we also note that the cost of evaluating only 100 sum-

maries for each system is relatively low (around 2 USD per system using ChatGPT). Since LLMs also conduct evaluations much faster than humans (around 2 minutes for LLMs versus 10 hours for human for 100 summaries), it may not pose significant barriers if one was to re-evaluate all compared systems on a single LLM.

**Limited Use of the Temporary Solution** Unfortunately, our temporary efficient framework doesn't apply to the relevance dimension, where the $R_i$ has no significant correlation with the performances of either RTS or MCQ. Moreover, the $r$ value may be dataset-dependent, and it is hard to decide where to draw this line. We leave for future work of developing better methods to gauge the reliability of LLM evaluations.

## Acknowledgements

Yang You is being sponsored by NUS startup grant (Presidential Young Professorship), Singapore MOE Tier-1 grant, ByteDance grant, ARCTIC grant, SMI grant and Alibaba grant.

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

> Score the following Summary given the corresponding Article with respect to consistency from one to five, where one indicates "inconsistency" and five indicates "perfect consistency". Note that consistency measures the factual alignment between the Summary and the Article, whether the Summary is faithful to the Article without introducing contradictions or misleading representations.
>
> Article: {article}
>
> Summary: {summary}
>
> Provide your reason in one sentence, then give a final score:

Table 8: Example prompt for the **RTS** method on the **consistency** dimension. Text in {blue}: the specific article and the corresponding summary to be evaluated.

> Score the following Summary given the corresponding Article with respect to fluency from one to five, where one indicates "disfluency" and five indicates "perfect fluency". Note that fluency measures the quality of individual sentences in the Summary, whether the Summary is well-written, grammatically correct, and readable on the sentence level.
>
> Article: {article}
>
> Summary: {summary}
>
> Provide your reason in one sentence, then give a final score:

Table 9: Example prompt for the **RTS** method on the **fluency** dimension. Text in {cyan}: the specific article and the corresponding summary to be evaluated.

> Score the following Summary given the corresponding Article with respect to coherence from one to five, where one indicates "incoherence" and five indicates "perfect coherence". Note that coherence measures the collective quality of the Summary, whether the Summary presents information that flows smoothly and avoids abrupt transitions or disjoint statements.
>
> Article: {article}
>
> Summary: {summary}
>
> Provide your reason in one sentence, then give a final score:

Table 10: Example prompt for the **RTS** method on the **coherence** dimension. Text in {cyan}: the specific article and the corresponding summary to be evaluated.

## A Evaluation Dimensions

Fabbri et al. (2021) has defined 4 evaluation dimensions as follows:

1. **Coherence**: The collective quality of all sentences. The summary should be well-structured and well-organized. The summary should not just be a heap of related information but should build from sentence to sentence to a coherent body of information about a topic.

> Choose an option from A to E in order to score the following Summary given the corresponding Article with respect to consistency from one to five, where one indicates "inconsistency" and five indicates "perfect consistency". Note that consistency measures the factual alignment between the Summary and the Article, whether the Summary is faithful to the Article without introducing contradictions or misleading representations.
>
> Article: {article}
>
> Summary: {summary}
>
> A: The Summary is totally inconsistent with the Article. Score: One.
> B: The majority of the Summary is inconsistent with the Article. Score: Two.
> C: Some information in the Summary is consistent with the Article whereas some are not. Score: Three.
> D: The majority of the Summary is consistent with the Article. Score: Four.
> E: All information included in the Summary is consistent with the Article. Score: Five.
>
> Your Answer (enter 1 letter from A to E):

Table 11: Example prompt for the **MCQ** method on the **consistency** dimension. Text in {cyan}: the specific article and the corresponding summary to be evaluated.

2. **Consistency**: The factual alignment between the summary and the summarized source. A factually consistent summary contains only statements that are entailed by the source document.

3. **Fluency**: The quality of individual sentences. Sentences in the summary should have no formatting problems, capitalization errors or obviously ungrammatical sentences (e.g., fragments, missing components) that make the text difficult to read.

4. **Relevance**: Selection of important content from the source. The summary should include only important information from the source document.

We follow the above definitions for designing ChatGPT's evaluation prompts.

Choose an option from A to E in order to score the following Summary given the corresponding Article with respect to fluency from one to five, where one indicates "disfluency" and five indicates "perfect fluency". Note that fluency measures the quality of individual sentences in the Summary, whether the Summary is well-written, grammatically correct, and readable on the sentence level.

Article: {article}

Summary: {summary}

A: The Summary is totally disfluent. Score: One.
B: The majority of the Summary is disfluent. Score: Two.
C: Some sentences in the Summary are fluent whereas some are not. Score: Three.
D: The majority of the Summary is fluent. Score: Four
E: All sentences in the Summary are fluent. Score: Five.

Your Answer (enter 1 letter from A to E):

Table 12: Example prompt for the **MCQ** method on the **fluency** dimension. Text in {cyan}: the specific article and the corresponding summary to be evaluated.

Choose an option from A to E in order to score the following Summary given the corresponding Article with respect to coherence from one to five, where one indicates "incoherence" and five indicates "perfect coherence". Note that coherence measures the collective quality of the Summary, whether the Summary presents information that flows smoothly and avoids abrupt transitions or disjoint statements.

Article: {article}

Summary: {summary}

A: The Summary is completely incoherent. Score: One.
B: The Summary is mostly incoherent. Score: Two.
C: The Summary is somewhat coherent. Score: Three.
D: The Summary is mostly coherent. Score: Four.
E: The Summary is completely coherent. Score: Five.

Your Answer (enter 1 letter from A to E):

Table 13: Example prompt for the **MCQ** method on the **coherence** dimension. Text in {cyan}: the specific article and the corresponding summary to be evaluated.

Choose a more consistent summary from Summary #1 and Summary #2 with respect to the corresponding Article by choosing an option from A, B, or C. Note that consistency measures the factual alignment between the summary and the Article, whether the summary is faithful to the Article without introducing contradictions or misleading representations.

Article: {article}

Summary #1: {summary from model A}

Summary #2: {summary from model B}

A: Summary #1 is more consistent.
B: Summary #2 is more consistent.
C: Both Summary #1 and Summary #2 are equally consistent.

Your choice (enter 1 letter from A to C):

Table 14: Example prompt for the **H2H** method on the **consistency** dimension. Text in {cyan}: the specific article, and the corresponding summaries generated by a pair of compared models.

Choose a more fluent summary from Summary #1 and Summary #2 with respect to the corresponding Article by choosing an option from A, B, or C. Note that fluency measures the quality of individual sentences in the summary, whether the summary is well-written, grammatically correct, and readable on the sentence level.

Article: {article}

Summary #1: {summary from model A}

Summary #2: {summary from model B}

A: Summary #1 is more fluent.
B: Summary #2 is more fluent.
C: Both Summary #1 and Summary #2 are equally fluent.

Your choice (enter 1 letter from A to C):

Table 15: Example prompt for the **H2H** method on the **fluency** dimension. Text in {cyan}: the specific article, and the corresponding summaries generated by a pair of compared models.

## B  Prompt Details and Design

We show the RTS prompts for relevance, consistency, fluency, and coherence in Table 1, Table 8, Table 9, and Table 10 respectively.

We show the MCQ prompts for relevance, consistency, fluency, and coherence in Table 2, Table 11, Table 12, and Table 13 respectively.

We show the H2H prompts for relevance, consistency, fluency, and coherence in Table 3, Table 14, Table 15, and Table 16 respectively.

To determine the exact definitions used in our prompts for each dimension, we re-use the first sentence from Fabbri et al. (2021)'s definition. We then prompt the LLM to provide a definition for the evaluated dimension, such as "define the word relevance in the context of summarization", then extract the key phrases generated that we believe to fit the definitions of Fabbri et al. (2021) to make up the full definition. We believe this approach may

Choose a more coherent summary from Summary #1 and Summary #2 with respect to the corresponding Article by choosing an option from A, B, or C. Note that coherence measures the collective quality of the summary, whether the summary presents information that flows smoothly and avoids abrupt transitions or disjoint statements.

Article: {article}

Summary #1: {summary from model A}

Summary #2: {summary from model B}

A: Summary #1 is more coherent.
B: Summary #2 is more coherent.
C: Both Summary #1 and Summary #2 are equally coherent.

Your choice (enter 1 letter from A to C):

Table 16: Example prompt for the **H2H** method on the **coherence** dimension. Text in {cyan}: the specific article, and the corresponding summaries generated by a pair of compared models.

| | Coh | Con | Flu | Rel | Avg |
|---|---|---|---|---|---|
| ChatGPT-RTS2 | 5 | 6 | 7 | 6 | 6.00 |
| ChatGPT-MCQ2 | 4 | 7 | 6 | 4 | 5.25 |
| ChatGPT-StarEval | 7 | 7 | 7 | 6 | 6.75 |

Table 17: Total correct pairs for alternative prompts. Coh: coherence; Con: consistency; Flu: fluency; Rel: relevance.

help the LLM to better evaluate the summaries according to definitions partially generated in its own language. Nevertheless, we didn't invest extensive efforts in prompt designs as this is not our key focus. We also demonstrate that our prompts have better evaluation results than two alternative prompts in Appendix C.

## C    Alternative prompts

We also use ChatGPT to evaluate with the exact prompts from Wang et al. (2023a). We name these prompts "StarEval" since they prompt the LLM to give one to five stars for the summary. In addition, we use ChatGPT to evaluate with alternative prompts for RTS and MCQ by using the full definition as shown in Appendix A instead of supplementing the definition with ChatGPT-generated phrases. We name these two prompts RTS2 and MCQ2 respectively.

We show the results of these alternative prompts in Table 17 and Table 18.

## D    Llama 2 Results

We report the results of using three different sizes of Llama 2 models as LLM evaluators in Table 19. As shown, while the smallest model (7B) exhibits very low correlations with human scores (and only significant on the consistency and relevance dimensions), the larger models (13B and 70B) demonstrate significant correlations with human scores on the full dataset level. However, even the best-performing 70B model fails to outperform the human correlation of BARTScore, and is completely overwhelmed by the results of ChatGPT and GPT-4. This suggests that the open-sourced Llama 2 models are not suitable to be used as zero-shot evaluators. Moreover, all Llama 2 models exhibit significant meta-correlations for at least one dimension.

| | | Coherence | | | Consistency | | | Fluency | | | Relevance | | |
|---|---|---|---|---|---|---|---|---|---|---|---|---|---|---|
| | | Spear. | Pear. | Kend. | Spear. | Pear. | Kend. | Spear. | Pear. | Kend. | Spear. | Pear. | Kend. |
| RTS2 | Human-Corr | 0.339 | 0.338 | 0.285 | 0.393 | 0.497 | 0.350 | 0.290 | 0.280 | 0.252 | 0.440 | 0.441 | 0.351 |
| | Meta-Corr | -0.559 | -0.476 | -0.424 | -0.748 | -0.843 | -0.576 | -0.825 | -0.823 | -0.636 | -0.385 | -0.506 | -0.212 |
| MCQ2 | Human-Corr | 0.430 | 0.423 | 0.355 | 0.327 | 0.483 | 0.306 | 0.258 | 0.396 | 0.229 | 0.240 | 0.258 | 0.206 |
| | Meta-Corr | -0.308 | -0.275 | -0.212 | -0.545 | -0.446 | -0.364 | -0.811 | -0.615 | -0.636 | -0.217 | -0.707 | -0.182 |
| StarEval | Human-Corr | 0.418 | 0.417 | 0.341 | 0.297 | 0.421 | 0.264 | 0.246 | 0.323 | 0.217 | 0.393 | 0.405 | 0.323 |
| | Meta-Corr | -0.084 | -0.101 | 0.000 | -0.664 | -0.684 | -0.545 | -0.441 | -0.460 | -0.212 | -0.497 | -0.579 | -0.303 |

Table 18: Results of using alternative prompt with ChatGPT. Light gray values are insignificant (p-value $\geq$ 0.05). Human-Corr reports the overall correlation of ChatGPT scores with human scores. Meta-corr shows the meta-correlation.

| | | Coherence | | | Consistency | | | Fluency | | | Relevance | | |
|---|---|---|---|---|---|---|---|---|---|---|---|---|---|---|
| | | Spear. | Pear. | Kend. | Spear. | Pear. | Kend. | Spear. | Pear. | Kend. | Spear. | Pear. | Kend. |
| 7B | Human-Corr | -0.001 | -0.000 | -0.001 | 0.114 | 0.130 | 0.104 | 0.043 | 0.038 | 0.039 | 0.067 | 0.064 | 0.057 |
| | Meta-Corr | 0.245 | -0.258 | 0.182 | -0.524 | -0.720 | -0.424 | -0.042 | 0.463 | -0.03 | 0.238 | 0.169 | 0.212 |
| 13B | Human-Corr | 0.153 | 0.165 | 0.123 | 0.180 | 0.209 | 0.162 | 0.187 | 0.179 | 0.167 | 0.234 | 0.266 | 0.192 |
| | Meta-Corr | -0.287 | -0.425 | -0.182 | -0.580 | -0.495 | -0.424 | -0.455 | -0.233 | -0.303 | -0.049 | -0.406 | 0.000 |
| 70B | Human-Corr | 0.234 | 0.254 | 0.186 | 0.357 | 0.395 | 0.319 | 0.155 | 0.161 | 0.134 | 0.248 | 0.285 | 0.200 |
| | Meta-Corr | -0.420 | -0.407 | -0.273 | -0.811 | -0.690 | -0.667 | -0.322 | -0.319 | -0.182 | -0.238 | -0.123 | -0.182 |

Table 19: Results of Llama 2 models of 7B, 13B, and 70B RTS correlations. Light gray values are insignificant (p-value $\geq$ 0.05). Human-Corr reports the overall correlation of LLM scores with human scores. Meta-corr shows the meta-correlation.

| ID | Coh | Con | Flu | Rel |
|---|---|---|---|---|
| M8 | 0.58 | 0.09 | 0.13 | 0.51 |
| M9 | 0.65 | 0.15 | 0.34 | 0.54 |
| M10 | 0.58 | 0.22 | 0.20 | 0.54 |
| M11 | 0.59 | 0.34 | 0.40 | 0.49 |
| M12 | 0.65 | 0.03 | 0.14 | 0.54 |
| M13 | 0.62 | 0.07 | 0.14 | 0.51 |
| M14 | 0.65 | 0.09 | 0.18 | 0.54 |
| M15 | 0.60 | 0.04 | 0.13 | 0.53 |
| M17 | 0.58 | 0.05 | 0.08 | 0.52 |
| M20 | 0.48 | 0.29 | 0.24 | 0.49 |
| M22 | 0.58 | 0.02 | 0.14 | 0.54 |
| M23 | 0.58 | 0.05 | 0.12 | 0.54 |

Table 20: The standard deviation of human annotations across different summarization systems and evaluation dimensions.

| ID | Model Name | Coh | Con | Flu | Rel | Avg |
|---|---|---|---|---|---|---|
| M22 | BART | 4.18 | 4.94 | 4.90 | 4.25 | 4.57 |
| M23 | Pegasus (C4) | 4.16 | 4.91 | 4.88 | 4.26 | 4.55 |
| M17 | T5 | 4.00 | 4.93 | 4.93 | 4.23 | 4.52 |
| M12 | Unified-ext-abs | 3.60 | 4.96 | 4.85 | 3.85 | 4.32 |
| M13 | ROUGESal | 3.44 | 4.82 | 4.86 | 3.83 | 4.24 |
| M15 | Closed book decoder | 3.35 | 4.95 | 4.80 | 3.67 | 4.19 |
| M14 | Multi-task (Ent + QG) | 3.20 | 4.90 | 4.74 | 3.63 | 4.12 |
| M8 | Pointer Generator | 3.29 | 4.65 | 4.79 | 3.55 | 4.07 |
| M9 | Fast-abs-rl | 2.38 | 4.67 | 4.50 | 3.52 | 3.77 |
| M10 | Bottom-Up | 2.73 | 4.25 | 4.42 | 3.38 | 3.70 |
| M20 | GPT-2 (zero-shot) | 3.63 | 3.40 | 3.97 | 3.30 | 3.58 |
| M11 | Improve-abs | 2.28 | 3.27 | 3.65 | 3.15 | 3.09 |

Table 21: The average human evaluation scores of various abstractive summarization models reported by Fabbri et al. (2021). We calculate the average (Avg) score of the reported coherence (Coh), consistency (Con), fluency (Flu), and relevance (Rel) scores. Rows are sorted according to the Avg column values in descending order.

# E Challenging Pairs

To count the total correct pairs, we only evaluate the challenging pairs, which consist of summarization systems of consecutive performances according to average human scores across all dimensions. Thus, each pair contains 2 summarization systems with the smallest difference in terms of average performance.

For instance, as shown in Table 21, M22 has the best average human score of 4.57, followed by M23 of 4.55, then M17 of 4.52. We thus compare model pairs of "M22-M23" and "M23-M17". The full challenge set is shown in Table 22.

For RTS, MCQ, and all other baseline metrics,

we simply need to compare the evaluated values across all systems, and each metric only needs to evaluate a total of 1200 summaries. However, for H2H, we need to evaluate a total of 6,600 summary pairs for the full standard set, and each pair needs to be evaluated twice with different summary positions (see § 3.1), resulting in a total of 13,200 LLM evaluations. Due to a limited budget, we thus only compare a challenge set of 11 pairs, reducing the total required LLM evaluations to 2,200.

| | | Coherence | | | Consistency | | | Fluency | | | Relevance | | |
|---|---|---|---|---|---|---|---|---|---|---|---|---|---|---|
| Model A | Model B | LLM | Human | | LLM | Human | | LLM | Human | | LLM | Human | |
| M22 | M23 | 65.5 | 53.5 | ✓ | 55.75 | 52.5 | ✓ | 61 | 49.5 | × | 58.75 | 49.5 | × |
| M23 | M17 | 48.25 | 52.5 | × | 47 | 49 | ✓ | 49 | 45.5 | ✓ | 45 | 52 | × |
| M17 | M12 | 44 | 66.5 | × | 43.25 | 48.5 | ✓ | 40.5 | 54.5 | × | 49.25 | 72.5 | × |
| M12 | M13 | 58 | 51 | ✓ | 56.5 | 54.5 | ✓ | 56.75 | 50 | × | 58 | 45 | × |
| M13 | M15 | 45.5 | 49.5 | ✓ | 52 | 46 | × | 48.75 | 52 | × | 51.25 | 60.5 | ✓ |
| M15 | M14 | 57 | 54 | ✓ | 55 | 53.5 | ✓ | 56.5 | 52 | ✓ | 54 | 57 | ✓ |
| M14 | M8 | 49.25 | 44 | ✓ | 50 | 54.5 | × | 47 | 46.5 | ✓ | 49.25 | 53.5 | × |
| M8 | M9 | 77.5 | 82 | ✓ | 78.5 | 53 | ✓ | 80.5 | 63.5 | ✓ | 76 | 54 | ✓ |
| M9 | M10 | 45 | 36 | ✓ | 41.5 | 58 | × | 46 | 44.5 | ✓ | 41.5 | 56 | × |
| M10 | M20 | 58.25 | 24 | × | 61.75 | 64 | ✓ | 63.75 | 61.5 | ✓ | 61.5 | 54.5 | ✓ |
| M20 | M11 | 56.5 | 82 | ✓ | 50 | 53 | × | 51 | 58.5 | ✓ | 50 | 53 | × |
| #CP | | | 8 | | | 7 | | | 7 | | | 4 | |

Table 22: #CP calculation for the **ChatGPT-H2H** metric of **Model A** over Model B. The numerical values in the middle section columns are aggregated scores for Model A. We omit the value for Model B, which is simply "100 - aggregated scores for Model A". We use "✓" to indicate both LLM and humans prefer the same model, and "×" otherwise. The model pairs are sorted in descending order according to the average human scores for each model.

| | Coherence | | | Consistency | | | Fluency | | | Relevance | | |
|---|---|---|---|---|---|---|---|---|---|---|---|---|
| ID | Spear. | Pear. | Kend. | Spear. | Pear. | Kend. | Spear. | Pear. | Kend. | Spear. | Pear. | Kend. |
| M8 | 0.420 | 0.383 | 0.323 | 0.229 | 0.273 | 0.209 | 0.274 | 0.245 | 0.236 | 0.519 | 0.509 | 0.438 |
| M9 | 0.174 | 0.243 | 0.142 | 0.119 | 0.265 | 0.103 | 0.256 | 0.258 | 0.219 | 0.254 | 0.307 | 0.200 |
| M10 | 0.365 | 0.415 | 0.292 | 0.305 | 0.452 | 0.251 | 0.258 | 0.288 | 0.223 | 0.367 | 0.378 | 0.284 |
| M11 | 0.378 | 0.420 | 0.320 | 0.404 | 0.488 | 0.335 | 0.227 | 0.288 | 0.182 | 0.501 | 0.511 | 0.394 |
| M12 | 0.208 | 0.237 | 0.160 | 0.087 | 0.020 | 0.082 | 0.086 | 0.107 | 0.071 | 0.438 | 0.451 | 0.354 |
| M13 | 0.455 | 0.473 | 0.359 | 0.178 | 0.037 | 0.167 | 0.063 | 0.151 | 0.055 | 0.403 | 0.403 | 0.329 |
| M14 | 0.433 | 0.467 | 0.355 | 0.114 | 0.187 | 0.105 | -0.007 | 0.055 | -0.005 | 0.421 | 0.435 | 0.336 |
| M15 | 0.372 | 0.385 | 0.279 | 0.189 | 0.316 | 0.177 | 0.100 | 0.087 | 0.085 | 0.252 | 0.288 | 0.193 |
| M17 | 0.291 | 0.320 | 0.233 | -0.086 | -0.061 | -0.084 | 0.017 | 0.046 | 0.015 | 0.204 | 0.199 | 0.175 |
| M20 | 0.382 | 0.394 | 0.310 | 0.528 | 0.501 | 0.430 | 0.367 | 0.349 | 0.307 | 0.292 | 0.278 | 0.237 |
| M22 | 0.215 | 0.293 | 0.162 | -0.072 | -0.052 | -0.070 | -0.114 | -0.131 | -0.101 | 0.201 | 0.300 | 0.172 |
| M23 | 0.392 | 0.427 | 0.320 | 0.003 | 0.305 | 0.002 | -0.078 | -0.022 | -0.069 | 0.223 | 0.324 | 0.184 |

Table 23: Spearman (Spear.) correlations, Pearson (Pear.) correlations, and Kendall's Tau (Kend.) between **ChatGPT-RTS** and human scores on the 100 summaries for each model. Values in light gray color are insignificant (p-value $\geq$ 0.05).

| | Coherence | | | Consistency | | | Fluency | | | Relevance | | |
|---|---|---|---|---|---|---|---|---|---|---|---|---|
| ID | Spear. | Pear. | Kend. | Spear. | Pear. | Kend. | Spear. | Pear. | Kend. | Spear. | Pear. | Kend. |
| M8 | 0.289 | 0.310 | 0.236 | 0.235 | 0.362 | 0.226 | 0.348 | 0.348 | 0.321 | 0.349 | 0.419 | 0.302 |
| M9 | 0.170 | 0.170 | 0.148 | 0.211 | 0.321 | 0.200 | 0.198 | 0.299 | 0.174 | 0.318 | 0.356 | 0.276 |
| M10 | 0.352 | 0.314 | 0.293 | 0.138 | 0.273 | 0.125 | 0.155 | 0.210 | 0.138 | 0.420 | 0.427 | 0.362 |
| M11 | 0.285 | 0.312 | 0.250 | 0.380 | 0.370 | 0.317 | 0.200 | 0.218 | 0.163 | 0.397 | 0.428 | 0.334 |
| M12 | 0.306 | 0.304 | 0.258 | -0.025 | -0.059 | -0.024 | 0.256 | 0.306 | 0.239 | 0.283 | 0.287 | 0.246 |
| M13 | 0.425 | 0.425 | 0.351 | 0.471 | 0.312 | 0.457 | 0.199 | 0.214 | 0.186 | 0.435 | 0.402 | 0.375 |
| M14 | 0.490 | 0.472 | 0.422 | 0.112 | 0.157 | 0.110 | 0.084 | 0.143 | 0.078 | 0.326 | 0.329 | 0.284 |
| M15 | 0.317 | 0.298 | 0.250 | 0.061 | 0.513 | 0.060 | 0.055 | 0.025 | 0.050 | 0.433 | 0.420 | 0.378 |
| M17 | 0.250 | 0.255 | 0.215 | -0.106 | -0.081 | -0.105 | -0.011 | 0.024 | -0.011 | 0.293 | 0.285 | 0.260 |
| M20 | 0.463 | 0.450 | 0.381 | 0.455 | 0.442 | 0.371 | 0.494 | 0.450 | 0.404 | 0.326 | 0.334 | 0.264 |
| M22 | 0.211 | 0.173 | 0.182 | -0.096 | -0.080 | -0.095 | -0.092 | -0.056 | -0.087 | 0.352 | 0.371 | 0.313 |
| M23 | 0.218 | 0.209 | 0.189 | 0.107 | 0.448 | 0.104 | -0.275 | -0.269 | -0.261 | 0.147 | 0.187 | 0.129 |

Table 24: Spearman (Spear.) correlations, Pearson (Pear.) correlations, and Kendall's Tau (Kend.) between **ChatGPT-MCQ** and human scores on the 100 summaries for each model. Values in light gray color are insignificant (p-value $\geq$ 0.05).

# F  Average ChatGPT scores

We present the average ChatGPT evaluation scores for each model across all dimensions in Table 25. Generally, the same trend holds for the individual systems, that ChatGPT score systems much more conservatively with RTS, and becomes more optimistic with MCQ.

| | Coherence | | | | Consistency | | | | Fluency | | | | Relevance | | | |
|---|---|---|---|---|---|---|---|---|---|---|---|---|---|---|---|---|
| ID | Chat-RTS | Chat-MCQ | GPT-4 | human | Chat-RTS | Chat-MCQ | GPT-4 | human | Chat-RTS | Chat-MCQ | GPT-4 | human | Chat-RTS | Chat-MCQ | GPT-4 | human |
| M8 | 2.43 | 3.70 | 4.56 | 3.29 | 4.21 | 4.70 | 4.77 | 4.65 | 2.51 | 3.76 | 4.61 | 4.79 | 3.56 | 4.33 | 4.71 | 3.55 |
| M9 | 1.80 | 3.51 | 4.41 | 2.38 | 3.93 | 4.69 | 4.94 | 4.67 | 2.16 | 3.45 | 4.15 | 4.50 | 3.51 | 4.33 | 4.82 | 3.52 |
| M10 | 2.05 | 3.49 | 4.05 | 2.73 | 3.77 | 4.52 | 4.60 | 4.25 | 2.23 | 3.43 | 3.92 | 4.42 | 3.45 | 4.25 | 4.62 | 3.38 |
| M11 | 1.70 | 2.63 | 2.93 | 2.28 | 2.36 | 4.11 | 3.72 | 3.27 | 1.71 | 2.66 | 2.77 | 3.65 | 2.92 | 4.06 | 3.87 | 3.15 |
| M12 | 2.75 | 3.92 | 4.75 | 3.60 | 4.46 | 4.75 | 5.00 | 4.96 | 2.59 | 3.84 | 4.72 | 4.85 | 3.89 | 4.40 | 4.95 | 3.85 |
| M13 | 2.91 | 3.99 | 4.76 | 3.44 | 4.40 | 4.73 | 4.89 | 4.82 | 2.69 | 3.93 | 4.69 | 4.86 | 3.90 | 4.41 | 4.85 | 3.83 |
| M14 | 2.38 | 3.87 | 4.54 | 3.20 | 4.25 | 4.70 | 4.99 | 4.90 | 2.42 | 3.83 | 4.53 | 4.74 | 3.67 | 4.32 | 4.84 | 3.63 |
| M15 | 2.65 | 3.81 | 4.58 | 3.35 | 4.33 | 4.75 | 4.90 | 4.95 | 2.60 | 3.84 | 4.61 | 4.80 | 3.78 | 4.32 | 4.82 | 3.67 |
| M17 | 3.05 | 4.11 | 4.89 | 4.00 | 4.84 | 4.87 | 4.97 | 4.93 | 3.29 | 4.08 | 4.79 | 4.93 | 4.30 | 4.43 | 4.97 | 4.23 |
| M20 | 2.00 | 2.95 | 2.99 | 3.63 | 2.66 | 3.57 | 3.51 | 3.40 | 2.22 | 3.02 | 3.05 | 3.97 | 2.73 | 3.78 | 3.47 | 3.30 |
| M22 | 3.54 | 4.09 | 4.92 | 4.18 | 4.83 | 4.82 | 4.97 | 4.94 | 3.57 | 4.06 | 4.94 | 4.90 | 4.41 | 4.43 | 4.95 | 4.25 |
| M23 | 3.00 | 4.14 | 4.88 | 4.16 | 4.74 | 4.84 | 4.94 | 4.91 | 3.08 | 4.05 | 4.76 | 4.88 | 4.10 | 4.45 | 4.96 | 4.26 |
| avg | 2.52 | 3.68 | 4.35 | 3.35 | 4.06 | 4.59 | 4.68 | 4.55 | 2.59 | 3.66 | 4.29 | 4.61 | 3.68 | 4.29 | 4.65 | 3.72 |

Table 25: Average scores for ChatGPT using RTS (Chat-RTS) and MCQ (Chat-MCQ), GPT-4 using RTS (GPT-4), as well as the human scores.

| Dimension | Generation |
|---|---|
| Consistency | The summary is inconsistent with the article as it **omits important details** and **repeats a phrase multiple times**, resulting in a score of 1. |
| Consistency | The summary includes some relevant information but **lacks coherence** and **omits important details**, resulting in a score of 2.5 out of 5 for consistency. |
| Fluency | The summary is well-written and grammatically correct, but **lacks important details from the article such as the potential link between antibiotics and weight gain, and the warnings about over-prescription of antibiotics and the development of superbugs.** Score: 3/5. |
| Fluency | The summary **lacks coherence** and **does not provide a clear understanding of the article**, resulting in a score of 1 for disfluency. |

Table 26: Examples of wrong reasons generated during RTS by ChatGPT that do not correspond to the evaluated dimension. **Bolded**: reasons that don't match the evaluated dimension.

| Dimension | Generation |
|---|---|
| Coherence | The summary is somewhat coherent but **includes incorrect information, such as stating that Merson restarted his row with Burnley instead of Andros Townsend, and does not flow smoothly;** therefore, I would give it a score of 3. |
| Consistency | The summary is consistent with the article as it mentions the helicopter airlifting patients to the hospital, but it **leaves out important details such as the number of people involved and their injuries.** Final score: 3. |
| Consistency | The summary accurately reflects the main points of the article, **although it could have mentioned Falcao's unhappiness in France**. Final score: 4. |
| Fluency | The summary is coherent and grammatically correct, but it **does not provide enough information about the incident**. Final score: 3. |
| Fluency | The summary is **only one sentence and do not provide enough information about the article's content.** Final score: 1. |

Table 27: Examples of wrong reasons generated during RTS by GPT-4 that don't correspond to the evaluated dimension. **Bolded**: reasons that don't match the evaluated dimension.

## G   Incorrect Reasons in RTS

We illustrate some of the observed incorrect reasons generated with the RTS method by ChatGPT that do not correspond to the evaluated dimension in Table 26. We further provide incorrect reasons generated by GPT-4 in Table 27.

We haven't compiled the exact number of mismatched reasons for either LLM, but we do observe a non-trivial fraction of the evaluations containing unrelated-to-dimension reasoning during preliminary verification. For instance, on the consistency dimension, ChatGPT-RTS has 42% evaluations containing dimension-irrelevant reasoning for the M11 (avg. score of 2.36) model, and 10% for the M17 model (avg. score of 4.84).

## H   GPT-4 Evaluator

We also look into the reasoning of GPT-4 and discover that it makes the same mistakes as ChatGPT by penalizing the summary for reasons unrelated to the evaluated dimension (see Table 27).

Another major difference is that GPT-4 tends to give overly generous scores. In one exceptionally extreme case, GPT-4 gives full scores for all generations by M12 in terms of consistency. Table 25 also shows the much higher average scores given by GPT-4 across all dimensions than those of ChatGPT-RTS.

| | Coherence | | | Consistency | | | Fluency | | | Relevance | | |
|---|---|---|---|---|---|---|---|---|---|---|---|---|
| ID | Spear. | Pear. | Kend. | Spear. | Pear. | Kend. | Spear. | Pear. | Kend. | Spear. | Pear. | Kend. |
| M8 | 0.429 | 0.446 | 0.362 | 0.449 | 0.597 | 0.433 | 0.412 | 0.409 | 0.385 | 0.425 | 0.489 | 0.365 |
| M9 | 0.288 | 0.346 | 0.243 | 0.349 | 0.350 | 0.331 | 0.462 | 0.465 | 0.407 | 0.413 | 0.490 | 0.358 |
| M10 | 0.495 | 0.480 | 0.403 | 0.518 | 0.683 | 0.469 | 0.406 | 0.579 | 0.360 | 0.510 | 0.537 | 0.429 |
| M11 | 0.584 | 0.572 | 0.474 | 0.529 | 0.536 | 0.439 | 0.448 | 0.436 | 0.358 | 0.500 | 0.519 | 0.405 |
| M12 | 0.245 | 0.360 | 0.209 | - | - | - | 0.387 | 0.506 | 0.372 | 0.169 | 0.106 | 0.149 |
| M13 | 0.271 | 0.271 | 0.230 | 0.535 | 0.444 | 0.518 | 0.093 | 0.337 | 0.089 | 0.237 | 0.337 | 0.206 |
| M14 | 0.263 | 0.312 | 0.222 | -0.040 | -0.026 | -0.040 | 0.311 | 0.432 | 0.291 | 0.277 | 0.315 | 0.241 |
| M15 | 0.403 | 0.418 | 0.342 | 0.158 | 0.504 | 0.155 | 0.240 | 0.242 | 0.225 | 0.307 | 0.402 | 0.267 |
| M17 | 0.285 | 0.240 | 0.246 | 0.390 | 0.748 | 0.385 | 0.091 | 0.131 | 0.088 | 0.210 | 0.253 | 0.185 |
| M20 | 0.427 | 0.418 | 0.334 | 0.378 | 0.362 | 0.313 | 0.482 | 0.468 | 0.402 | 0.479 | 0.491 | 0.391 |
| M22 | 0.026 | 0.004 | 0.023 | 0.346 | 0.608 | 0.343 | 0.248 | 0.141 | 0.242 | 0.143 | 0.084 | 0.126 |
| M23 | 0.320 | 0.330 | 0.283 | 0.360 | 0.168 | 0.354 | 0.267 | 0.136 | 0.251 | 0.005 | 0.003 | 0.005 |

Table 28: Spearman (Spear.) correlations, Pearson (Pear.) correlations, and Kendall's Tau (Kend.) between **GPT-4 RTS** and human scores on the 100 summaries for each model. Values in light gray color are insignificant (p-value $\geq 0.05$). Note that for the consistency of M12, correlations cannot be calculated because GPT-4 gives 5 scores to all examples.

| | Coherence | | | Consistency | | | Fluency | | | Relevance | | |
|---|---|---|---|---|---|---|---|---|---|---|---|---|
| ID | Spear. | Pear. | Kend. | Spear. | Pear. | Kend. | Spear. | Pear. | Kend. | Spear. | Pear. | Kend. |
| M8 | 0.464 | 0.455 | 0.407 | 0.557 | 0.613 | 0.526 | 0.391 | 0.367 | 0.341 | 0.517 | 0.514 | 0.471 |
| M9 | 0.436 | 0.415 | 0.416 | 0.215 | 0.342 | 0.195 | 0.578 | 0.571 | 0.517 | 0.450 | 0.485 | 0.412 |
| M10 | 0.374 | 0.345 | 0.340 | 0.300 | 0.297 | 0.270 | 0.631 | 0.585 | 0.551 | 0.506 | 0.507 | 0.459 |
| M11 | 0.326 | 0.428 | 0.307 | 0.437 | 0.421 | 0.393 | 0.354 | 0.509 | 0.303 | 0.467 | 0.486 | 0.419 |
| M12 | 0.500 | 0.469 | 0.444 | 0.446 | 0.486 | 0.424 | 0.273 | 0.320 | 0.241 | 0.588 | 0.591 | 0.542 |
| M13 | 0.462 | 0.450 | 0.408 | 0.470 | 0.510 | 0.445 | 0.325 | 0.345 | 0.282 | 0.515 | 0.498 | 0.475 |
| M14 | 0.524 | 0.460 | 0.463 | 0.325 | 0.293 | 0.307 | 0.216 | 0.257 | 0.194 | 0.501 | 0.476 | 0.455 |
| M15 | 0.534 | 0.499 | 0.470 | 0.266 | 0.326 | 0.251 | 0.383 | 0.369 | 0.337 | 0.584 | 0.550 | 0.535 |
| M17 | 0.312 | 0.294 | 0.274 | -0.016 | -0.008 | -0.015 | 0.251 | 0.261 | 0.225 | 0.530 | 0.479 | 0.504 |
| M20 | 0.636 | 0.623 | 0.566 | 0.733 | 0.664 | 0.644 | 0.566 | 0.556 | 0.492 | 0.570 | 0.565 | 0.496 |
| M22 | 0.314 | 0.338 | 0.276 | 0.245 | 0.151 | 0.241 | 0.371 | 0.341 | 0.330 | 0.471 | 0.412 | 0.451 |
| M23 | 0.380 | 0.386 | 0.337 | 0.158 | 0.496 | 0.153 | 0.296 | 0.275 | 0.259 | 0.450 | 0.505 | 0.421 |

Table 29: $R_i$, the reliability indicator calculated by the Spearman (Spear.) correlations, Pearson (Pear.) correlations, and Kendall's Tau (Kend.) between ChatGPT-RTS and ChatGPT-MCQ. Values in light gray color are insignificant (p-value $\geq 0.05$).