# OpenReview forum: "Large Language Models are Not Yet Human-Level Evaluators for Abstractive Summarization"
_EMNLP/2023/Conference — EMNLP 2023 Findings_

### Official Review · Reviewer_i7an · 2023-08-03

**Soundness:** 3

**Excitement:**

3: Ambivalent: It has merits (e.g., it reports state-of-the-art results, the idea is nice), but there are key weaknesses (e.g., it describes incremental work), and it can significantly benefit from another round of revision. However, I won't object to accepting it if my co-reviewers champion it.

**Paper Topic And Main Contributions:**

This paper tests the evaluation ability of ChatGPT and GPT-4 on the SummEval dataset with different assessment methods and prompt designs (RTS, MCQ, and H2H). The analysis shows ChatGPT correlates with human judgments better than commonly used evaluation metrics but they may not be reliable because there are some drawbacks, such as that they are candidate-dependant and dimension-dependant. Finally, a temporary efficient framework that uses the correlation between the results of RTS and MCQ as an indicator is proposed.

**Questions For The Authors:**

- Question A: Why the extractive systems in the SummEval dataset are excluded? I do not think it is necessary.

- Question B: What are the distributions of human scores for the examples corresponding to different systems and dimensions? For example, I want to see the statistics of the human scores for each system. This will affect my understanding of the LLMs' dependence on candidates and dimensions.

**Reasons To Accept:**

- This paper keeps pace with the newest advances in NLG evaluation, which leverages LLMs for human-like evaluation. Different from the studies that test LLMs on common datasets and claim their superiority, a more in-depth analysis is presented, and some limitations of LLMs-based evaluation are revealed.

- A meta-correlation metric is defined to measure the degree to which the performance of LLMs-based evaluators is affected by the quality of the evaluated systems, which can be used to further investigate the bias when using LLMs-based evaluators.

**Reasons To Reject:**

- This paper only uses one dataset for the experiment. More experiments on other datasets and LLMs are needed to support the title of this paper.

- The authors argue that ChatGPT-RTS generates false or unrelated-to-dimension reasoning and thereby penalizes the summaries according to such false premise. Though several cases are shown in the appendix, it is insufficient to draw such a conclusion. I would like to see a more detailed analysis of this problem and more human examinations of the correctness of LLMs' reasoning may be required.

- The paper does not tell us the extent to which the commonly used evaluation metrics and human annotators are candidate-dependant and dimension-dependant. Without these results, it is hard to comprehensively understand the limitations of LLM evaluators from a relative perspective. In other words, LLMs have these limitations, what about the previous evaluation metrics and human evaluators?

- The proposed efficient framework lacks experimental results and is hard to use in practice. It will be very hard for researchers to determine the reliability score $r$. This framework also seems to have the limitations of LLM evaluators stated in this paper.

**Reproducibility:**

4: Could mostly reproduce the results, but there may be some variation because of sample variance or minor variations in their interpretation of the protocol or method.

**Reviewer Confidence:**

4: Quite sure. I tried to check the important points carefully. It's unlikely, though conceivable, that I missed something that should affect my ratings.

**Typos Grammar Style And Presentation Improvements:**

Typos:

- line 114, line 457: "human judgment" $\rightarrow$ "human judgments"

- line 602: "are" $\rightarrow$ "is"

---

> ### Author Rebuttal · Authors · 2023-08-29
>
> We thank the reviewer for the thoughtful review. Let us clarify and address the raised concerns as follows:
>
> * _Request for experiments on more datasets and LLMs_
>
> Kindly refer to our response for Reviewer iAX6. We will include these results in the Appendix as they are __consistent__ with the conclusions of our paper.
>
> * _Concerns on the "conclusion" that RTS penalizes summaries according to false premises_
>
> This is actually not a conclusion but __one plausible reason__ why RTS is giving consistently lower scores than MCQ (lines 486-487, Appendix Table 23). We haven't tabulated the results, but we do observe a __non-trivial fraction__ of the annotations with unrelated-to-dimension reasoning. For instance, on the consistency dimension, M11 (scored 2.36) has __42%__ reasonings containing dimension-irrelevant reasonings, and even M17 with a top score of 4.84 has __10%__ such reasonings.
>
> * _Whether previous metrics and human annotations are candidate and dimension dependent_
>
> As we define candidate-dependence and dimension-dependence as whether the metric exhibits consistent degrees of __human alignment__, it doesn't apply to human annotations which are __used as the reference__. We provide the meta-correlations (Kendall tau) for all evaluated metrics in the table below (__bolded__: p < 0.05).
>
> |   metric    | Coherence | Consistency | Fluency | Relevance |
> | ----------- | ----------- | ----------- | ----------- | ----------- |
> | Rouge 1                        | 0.014  | -0.455 | -0.420 | -0.329 |
> | Rouge 2                        | __0.615__ | -0.217 | -0.203 | -0.084 |
> | Rouge L                        | 0.259  | -0.510 | -0.203 | -0.182 |
> | BERTScore.                  | -0.315 | __-0.580__| -0.455 | __-0.643__|
> | BARTScore                   |__-0.916__| -0.161 | 0.154  | __-0.769__|
> | BARTScore-CNN          |__-0.797__| -0.678 | -0.510 | __-0.825__|
> | BARTScore-CNN-PARA|__-0.769__| -0.692 | -0.231| __-0.853__ |
> | ChatGPT-RTS                | -0.189 | __-0.797__ | __-0.748__ | -0.559 |
> | GPT-4-RTS                    | -0.497 |  -0.364 | __-0.692__  | __-0.958__  |
>
>
> Rouge doesn't exhibit significant negative meta-correlation, whereas the neural metrics show significant meta-correlations for some dimensions, although varying from metric to metric. Nevertheless, although the previous neural metrics may equally suffer from dimensional and candidate dependencies, they are __frequently complemented with human evaluation__; however, one may be tempted to replace human evaluation with LLM Evaluators given the various recent strong results. Thus, our goal is to *raise awareness of the potential unreliability* of LLM evaluations out-of-the-box.
>
> * _Concerns on the reliability score, r_
>
> ‒ The score, r, is the correlation of P^{RTS} and P^{MCQ}. We believe the reviewer is asking if it is hard to know to what extent r indicates the LLM scores are reliable? For SummEval our observed threshold is around 0.5. Admittedly, the threshold is likely to be dataset-dependent, so we acknowledge the reviewer's concern about the limitation of this solution.
>
> ‒ Nevertheless, our main focus is to demonstrate that ChatGPT and GPT-4 are not yet human-level evaluators, and this mitigation method is only a __temporary solution__ that we explore. We will leave designing better metrics to future work (such as how to effectively utilize LLM evaluators).
>
> * _Concerns on the exclusion of extractive systems_
>
> Our focus is on abstractive summarization because it poses stronger challenges for evaluation due to its much wider answer space and open nature (lines 41-56). For instance, the extractive systems achieve close to full marks on consistency and fluency [Fabbri et al., 2021]. Thus, we only test the LLM evaluators on the relevant systems.
>
> * _Distribution of human scores for different systems and dimensions_
>
> We show the average scores for each system and dimension in Appendix Tab 23 (briefly referred to in lines 476-477). The SummEval paper presents the standard deviation for each dimension in Fig. 2 (rightmost figure), and we also provide a detailed per-model breakdown in the table below. Given the minimum step size for annotation is 1, these values are considered very small, indicating substantial annotator agreements.
>
> |   metric    | Coherence | Consistency | Fluency | Relevance |
> | ----------- | ----------- | ----------- | ----------- | ----------- |
> |M8 | 0.58 | 0.09 | 0.13 | 0.51|
> |M9 | 0.65 | 0.15 | 0.34 | 0.54|
> |M10 | 0.58 | 0.22 | 0.20 | 0.54|
> |M11 | 0.59 | 0.34 | 0.40 | 0.49|
> |M12 | 0.65 | 0.03 | 0.14 | 0.54|
> |M13 | 0.62 | 0.07 | 0.14 | 0.51|
> |M14 | 0.65 | 0.09 | 0.18 | 0.54|
> |M15 | 0.60 | 0.04 | 0.13 | 0.53|
> |M17 | 0.58 | 0.05 | 0.08 | 0.52|
> |M20 | 0.48 | 0.29 | 0.24 | 0.49|
> |M22 | 0.58 | 0.02 | 0.14 | 0.54|
> |M23 | 0.58 | 0.05 | 0.12 | 0.54|
>
> References:
>
> [Fabbri et al., 2021] SummEval: Re-evaluating Summarization Evaluation

---

### Official Review · Reviewer_iAX6 · 2023-08-05

**Soundness:** 3

**Excitement:**

4: Strong: This paper deepens the understanding of some phenomenon or lowers the barriers to an existing research direction.

**Paper Topic And Main Contributions:**

The paper conducts a comprehensive analysis to investigate the stability and reliability of LLMs as automatic evaluators for abstractive summarization and it has drawn many interesting conclusions.

**Questions For The Authors:**

a. Have you tried other datasets, and are the results on other datasets such as *newsroom* consistent with the conclusions of this paper?

b. Have you tried other LLMs, and are the results on other LLMs such as *LLaMA* consistent with the conclusions of this paper?

**Reasons To Accept:**

1. The experiments in this paper were conducted in great detail, accompanied by experimental results, case studies, and analysis of each viewpoint.

2. The experimental results and conclusions of the paper will be of help for researchers/users to choose proper and efficient evaluators for abstractive summarization.

3. The paper provides constructive opinions about building a temporary efficient framework as an evaluation pipeline to tell when LLM evaluators have become unreliable and employ further human evaluation when necessary.

**Reasons To Reject:**

1. The experimental results and conclusions are all based on one dataset *SummEval*. Therefore, these conclusions are not convincing and cannot be generalized to all abstractive summarization tasks. Of course, *SummEval* is basically the most challenging dataset for summary tasks.

2. The paper only selects a commercialized LLM chatGPT to conduct experiments, and it is necessary to explore the performance of open source LLMs.

**Reproducibility:**

4: Could mostly reproduce the results, but there may be some variation because of sample variance or minor variations in their interpretation of the protocol or method.

**Reviewer Confidence:**

4: Quite sure. I tried to check the important points carefully. It's unlikely, though conceivable, that I missed something that should affect my ratings.

---

> ### Author Rebuttal · Authors · 2023-08-29
>
> We thank the reviewer for the thoughtful review. Let us clarify and address the raised concerns as follows:
>
> * _Experiments on more datasets_
>
> We believe there is no other sufficiently large dataset that __as reliable and representative__ for abstractive summarization as SummEval. We feel NewsRoom has significant limitations:
>
> ‒ *Not representative*: Annotations of only 4 abstractive systems (3 are the same model with different training data), only 60 examples, making it much more suspectable to out-liers
>
> ‒ *Outdated*: All 4 systems used are __before 2018__ with mediocre performance (human scores range from 2.40 - 3.73, on par with the worst-performing candidates in SummEval), thus not representative of the stronger recent summarization models.
>
> ‒ *Low quality*: The human evaluation of NewsRoom is done by __crowed-sourced annotators__, which are discovered to be questionable by SummEval as well as [Karpinska et al., 2021]
>
> We have run ChatGPT (gpt-3.5-turbo-0613) with NewsRoom, but observe very low correlations (Kendall tau) as detailed below ( __bold__ for p < 0.05):
> |   metrics                   | Coherence | Consistency | Fluency | Relevance |
> | -----------                 | -----------    | ----------- | ----------- | ----------- |
> | BARTScore              | __0.324__   | __0.431__  | __0.248__   | __0.422__ |
> |  ChatGPT  |      0.012     | __0.296__ | __0.122__ | __0.208__ |
>
> Broken down to individual systems, there is almost no correlation except for pointer n, which is trained on the full NewsRoom dataset.
> |   metrics                   | Coherence | Consistency | Fluency | Relevance |
> | -----------                 | -----------    | ----------- | ----------- | ----------- |
> | abstractive               |     0.029      |    -0.106    |    0.032   |   -0.019    |
> |  pointer c                  |      -0.101     |   0.020     |    0.109   |    0.019    |
> |  pointer n                  |      -0.047     | __0.346__ | __0.263__ |   0.134  |
> |  pointer s                  |      -0.041     |  -0.070      |   -0.013   |  -0.016   |
>
> Upon further analysis, the majority (83% on coherence, 70% on consistency, 75% on fluency, and 67% on relevance) of the summaries receive a score of 1.  We thus believe the overly mediocre candidates have led to low correlation scores by ChatGPT.
>
> * _Experiments on open-sourced LLMs_
>
> As discussed in our limitations (lines 618-620), our paper only seeks to explore __the best potential__ of LLMs as human-replacements, and thus select __the strongest models__ (i.e. ChatGPT and GPT-4 [OpenAI, 2023]).
> Nevertheless, we show results on all three sizes of "Llama-2-chat" models below in terms of Kendall tau (__bold__ indicates p < 0.05).
>
> |   metrics   | Coherence | Consistency | Fluency | Relevance |
> | ----------- | ----------- | ----------- | ----------- | ----------- |
> | correlation (7B)  | 0.016  |  __0.203__ | __0.074__ | __0.080__ |
> | correlation  (13B)  | __0.223__  |  __0.202__ | __0.117__ | __0.246__ |
> | correlation  (70B)  | __0.186__  |  __0.319__ | __0.136__ | __0.200__ |
> | meta-correlation  (7B)    | 0.112 | __-0.678__| -0.371 | -0.245|
> | meta-correlation   (13B)   | -0.392 | -0.497 | -0.049 | -0.420 |
> | meta-correlation   (70B)   | -0.378 | __-0.804__ | -0.315 | -0.238 |
>
> As shown, the Llama-2 cannot even outperform BARTScores in terms of dataset-level correlations. However, they also exhibit dimensional and candidate dependencies, which is __consistent__ with the conclusions of this paper.
>
> References:
>
> [Karpinska et al., 2021] The perils of using Mechanical Turk to evaluate open-ended text generation. (EMNLP 2021)
>
> [OpenAI, 2023] GPT-4 Technical Report

---

### Official Review · Reviewer_LdaQ · 2023-08-05

**Soundness:** 3

**Excitement:**

4: Strong: This paper deepens the understanding of some phenomenon or lowers the barriers to an existing research direction.

**Paper Topic And Main Contributions:**

Topic: Meta evaluation for (abstractive) summarization task. This paper tries to find the effectiveness of recent LLMs as a summarization metric (taking ChatGPT and GPT-4 as examples).
Paper describes how to use ChatGPT as a metric (in 3 variants), and tries to find strengths and weaknesses of them in comparison to alternative metrics: ROUGE (3 variants), BERTScore (1variant), BARTScore(3 variants). Findings are based on SummEval benchmark dataset containing human annotations for a bunch of abstractive summarization models. They show that ChatGPT outperforms all others. Then they ask “can we replace human annotators with ChatGPT”. Answer: No (or not yet). Then, what about GPT-4? Answer: Not yet.

*Contributions:*
1. Describes in detail how to use instruction finetuned LLMs such as ChatGPT as a metric. They formulate 3 flavors: reason-then-score (RTS), multiple-choice-question(MCQ), head-to-head(H2H)
1. Meta evaluation of metrics: authors show the strengths and weaknesses of ChatGPT variants and how they stack up against Rouge, BERTScore, BartScore. They show which flavor is the winner (spoiler: ChatGPT-RTS).
1. Then stability of LLM Evaluators: They define/propose a meta-correlation metric. Finding: they show ChatGPT become unreliable (i.e, hard time ranking summaries)
   1. when summarizatiion system qualities are too close (as per human scores).
   1. when summary qualities reach highest (correlation approach zero, indicating randomness)
1. They propose a method to detect when ChatGPT metric becomes unreliable so as to invoke human annotators as a fall back

**Questions For The Authors:**

* Q1) (Line 328) If there were 3 expert scores per item, did you just use simple average of 3?
  Any per-annotator normalization (such as z-score)?

* Q2) In table 8, is it meaningful to add \rho(P^{MCQ}, P^{RTS}) for a comparison?

**Reasons To Accept:**

Important problem/research question in the post-ChatGPT era.
Nice formulation of the problem and many contributions (See contributions).
Straightforward process to tackle the research problem.
Some useful analysis about when LLM as metric are reliable and when they are not (see Contributions).

**Reasons To Reject:**

#R1) A hypothesis in this work is that “LLM should work well regardless of the quality of the evaluated systems” (Line 299) needs careful consideration. This assumes that human annotations are perfect. But in practice human annotations are also biased (SummEval dataset is know to have poor inter-annotator agreement, as per Fabri et al 2021). In similar tasks such as machine translation (see WMT metrics shared task), there has been normalizations to account for the variance in human scores, which attempt to fix it. This paper doesn't not provide details on the topic, but instead assumes human scores are perfect.

#R2) This paper shows that LLM based metric becomes unreliable when the summarizer model quality becomes near human. When quality becomes near human, number of ties increase, and an importantly ranking correlation coefficients (such as the one used in this paper), does not properly reward ties.  There are many variants of Kendall Tau, attempting to handle ties in annotation, which would be worthwhile to explore (Authors have not disclosed what variant of Kendall's Tau formulation they used).

**Reproducibility:**

4: Could mostly reproduce the results, but there may be some variation because of sample variance or minor variations in their interpretation of the protocol or method.

**Reviewer Confidence:**

3: Pretty sure, but there's a chance I missed something. Although I have a good feel for this area in general, I did not carefully check the paper's details, e.g., the math, experimental design, or novelty.

**Typos Grammar Style And Presentation Improvements:**

* Presentation improvements: Tables appear to have too many numbers than needed to make a point. Given Kendall’s Tau is capturing ranking correlation, maybe we can drop Spearman coefficient? You may present all those numbers on the web. Also, some of the tables in Appendix maybe presented in website/gitrepo instead of appendix.

* Bibliography: A lot of references are from arXiv. Please reference peer-reviewed versions of the papers instead of drafts. Also, note that a lot of references are missing URLs (which can be easily spot blue color of paper title).
* Appendix: a lot of white space. Consider reordering tables and sections to minimize whitespace.
* Appendix: Tables 22+ should have same order of rows as in Table21 which ordered them by decreasing order of human performance.

---

> ### Author Rebuttal · Authors · 2023-08-29
>
> We thank the reviewer for the thoughtful and detailed review, with many constructive suggestions. Let us clarify and address the raised concerns as follows:
>
> * _We only use the __expert annotation portions__ of SummEval with __high__ inter-annotator agreement:_
>
> Although the __crowd-sourced annotator__ agreement is low, we use the **expert annotations** instead (as stated in line 328) with a kappa coefficient of 0.713 (discussed in the SummEval paper). The coefficient is super strong given the context of human evaluations. For instance, the highest kappa by [Chiang and Lee, 2023] was only 0.33. We thus use SummEval for its good annotation quality.
>
> * _We did not assume that human annotations are perfect and unbiased:_
>
> We believe that **no** human or LLM alike is perfect or unbiased. Instead, we explore whether LLM evaluators are __aligned with human experts__, and may naturally exhibit the same bias as a human would. In other words, we examine whether we can reliably replace human annotators with LLMs.
>
> * _How to combine the 3 scores_
>
> ‒ We used an average of the 3 expert scores, following several published works such as [Guan and Huang, 2020], [ChHun et al., 2022], and [Chiang and Lee, 2023].
>
> ‒ Following the reviewer's suggestion, we re-calculate the correlations using per-rate z-normalized human scores and observe the correlation for each model at each dimension exhibits __very little change__ (avg. &pm;0.005) from the results using non-normalized scores.  Furthermore, the meta-correlation for RTS with ChatGPT is __still significantly negative__ for consistency, fluency, and relevance at -0.797, -0.741, and -0.580. We thus suggest that z-normalization has __very little impact__ on SummEval human evaluation scores.
>
> * _Kendall Tau implementation_
>
> we used the "scipy.stats.kendalltau" functions' default settings, which uses the tau-b variant that __accounts for ties__. We will add this detail to the main paper.
>
> * _Whether it is meaningful to add \rho(P^{MCQ}, P^{RTS}) in Tab 8_
>
> ‒ We believe it is not necessary to include  \rho(P^{MCQ}, P^{RTS}), denoted as R_i.  We only intend to investigate __whether R_i is a good indicator of the performance__ of either RTS or MCQ, and we demonstrate so __by R_i's strong positive correlation__ with P^{MCQ} or P^{RTS} in certain dimensions.
>
> ‒ However, one may be interested in finding a threshold value of R_i for relatively good reliability of either P^{MCQ} and P^{RTS}. We observe that generally 0.5 is a good threshold for SummEval, and will include a detailed table for paired R_i values with P^{MCQ} and P^{RTS} in the paper.
>
> * _Regarding suggestions for references:_
>
> We acknowledge that some references are from arXiv, despite our best efforts to find their peer-reviewed versions. This is because the field of "LLM-Evaluators" is __relatively recent and actively growing__. Most of the works are in 2023 or the end of 2022 and do __not yet__ get enough chance to be published. We will carefully go through each reference again for updates and inclusion of URLs.
>
> * We would like to provide additional results using Llama-2 models, which are much weaker than ChatGPT and GPT-4, but also demonstrate dimensional and candidate dependences in terms of meta-correlation (__bolded__: p-value&leq;0.05)
>
> |   metrics   | Coherence | Consistency | Fluency | Relevance |
> | ----------- | ----------- | ----------- | ----------- | ----------- |
> | correlation (7B)  | 0.016  |  __0.203__ | __0.074__ | __0.080__ |
> | correlation  (13B)  | __0.223__  |  __0.202__ | __0.117__ | __0.246__ |
> | correlation  (70B)  | __0.186__  |  __0.319__ | __0.136__ | __0.200__ |
> | meta-correlation  (7B)    | 0.112 | __-0.678__| -0.371 | -0.245|
> | meta-correlation   (13B)   | -0.392 | -0.497 | -0.049 | -0.420 |
> | meta-correlation   (70B)   | -0.378 | __-0.804__ | -0.315 | -0.238 |
>
> References:
>
> [Chiang and Lee, 2023] Can Large Language Models Be an Alternative to Human Evaluation? (ACL 2023)
>
> [Guan and Huang, 2020] UNION: An Unreferenced Metric for Evaluating Open-ended Story Generation (EMNLP 2020)
>
> [ChHun et al., 2022] Of Human Criteria and Automatic Metrics: A Benchmark of the Evaluation of Story Generation (COLING 2022)

---

### Meta-Review · Area_Chair_ogVa · 2023-09-14

**Recommendation:** 3

**Metareview:**

The paper evaluates recent LLMs (Language Model Metrics) as summarization metrics, using ChatGPT and GPT-4. It provides an extensive exploration of ChatGPT's capabilities as a metric and juxtaposes its strengths and weaknesses against alternative metrics such as ROUGE, BERTScore, and BARTScore. These evaluations are grounded in the SummEval benchmark dataset, which includes human annotations for various summarization models.

While some reviewers expressed concerns about the use of a single dataset in the experiments, I believe it's important to note that this limitation is not a valid reason for rejection. This limitation arises due to the lack of alternative high-quality datasets for meta-evaluation in abstraction summarization.

However, reviewers did raise legitimate concerns regarding missing experiments involving other (commercial) LLMs, the need for further human examination of LLM reasoning correctness, and practical guidance on utilizing the efficient framework. The authors have provided additional results in their rebuttals, which should be incorporated into the paper along with a comprehensive discussion to address these concerns.

---

### Decision · Program_Chairs · 2023-10-07

**Decision:**

Accept-Findings

**Comment:**

The paper evaluates recent LLMs (Language Model Metrics) as summarization metrics, using ChatGPT and GPT-4. It provides an extensive exploration of ChatGPT's capabilities as a metric and juxtaposes its strengths and weaknesses against alternative metrics such as ROUGE, BERTScore, and BARTScore. These evaluations are grounded in the SummEval benchmark dataset, which includes human annotations for various summarization models.

While some reviewers expressed concerns about the use of a single dataset in the experiments, I believe it's important to note that this limitation is not a valid reason for rejection. This limitation arises due to the lack of alternative high-quality datasets for meta-evaluation in abstraction summarization.

However, reviewers did raise legitimate concerns regarding missing experiments involving other (commercial) LLMs, the need for further human examination of LLM reasoning correctness, and practical guidance on utilizing the efficient framework. The authors have provided additional results in their rebuttals, which should be incorporated into the paper along with a comprehensive discussion to address these concerns.